# Sparse Training of Neural Networks based on Multilevel Mirror Descent

## Abstract

We introduce a dynamic sparse training algorithm based on linearized Bregman iterations / mirror descent that exploits the naturally incurred sparsity by alternating between periods of static and dynamic sparsity pattern updates. The key idea is to combine sparsity-inducing Bregman iterations with adaptive freezing of the network structure to enable efficient exploration of the sparse parameter space while maintaining sparsity. We investigate convergence of the loss by embedding our method in a multilevel optimization framework. Under a suitable PŁ-type assumption on the loss we prove a general noise-neighborhood result as well as convergence to the minimal loss using a mixed exact/stochastic gradient setup. Furthermore, we empirically show that our algorithm can produce highly sparse and accurate models on standard benchmarks. We also show that the theoretical number of FLOPs compared to SGD training can be reduced from 38% for standard Bregman iterations to 6% for our method while maintaining test accuracy. We additionally show a training time reduction by about 50%, when using a sparsity-aware CPU implementation of our method.

## 1 Introduction

Deep neural networks have produced astonishing results in various areas such as computer vision and natural language processing (Mohd Noor & Ige, 2025) but demand significant memory and specialized hardware, contributing to growing concerns about their environmental impact, particularly the carbon footprint of training and inference (Dhar, 2020).

In response, researchers have explored techniques for developing compact and efficient models, such as sparse neural networks, wherein many neuron connections are absent. Within this context, the Lottery Ticket Hypothesis (Frankle & Carbin, 2019) plays a central role, suggesting that every dense network contains a sparse subnetwork that, when trained independently, can achieve comparable accuracy.

There are two main approaches to obtain sparse neural networks: pruning and sparse training. In pruning, a dense model is first trained and unwanted connections are removed afterward. Because this usually causes a drop in performance, the pruned model is often retrained with the sparsity pattern fixed. By contrast, sparse training incorporates mechanisms that encourage sparsity already during training. Pruning methods themselves vary widely, depending on how weights are selected for removal and at what stage of training pruning is applied. A key distinction is between unstructured pruning, which eliminates individual weights, and structured pruning, which removes entire components such as neurons or filters, see Hoefler et al. (2021) for an extensive overview.

In addition to pruning-based methods, sparsity can also be encouraged by including explicit regularization terms in the loss function. A common example is Lasso regularization, which uses the $\ell_1$-norm as a penalty in the objective function (Tibshirani, 1996). The resulting optimization problem can be solved using algorithms such as Proximal Gradient Descent (Rosasco et al., 2020; Mosci et al., 2010). A conceptually different approach is to enforce sparsity through *implicit regularization* which can be achieved using mirror descent (Nemirovskij & Yudin, 1983). Here we would like to highlight a series of works (Huang et al., 2016; Azizan et al., 2021; Bungert et al., 2021; 2022; Wang & Benning, 2023; Heeringa et al., 2025b) that utilize mirror descent or linearized Bregman iterations to induce sparsity in neural networks without explicit regularization.

Linearized Bregman iterations are equivalent to mirror descent, but they are typically formulated and analyzed with less regularity assumptions on the mirror map (e.g., compared to Azizan et al. (2021)), thereby lending themselves toward non-smooth sparsity-promoting mirror maps. This was exploited in (Bungert et al., 2022) to devise the LinBreg algorithm which essentially is a stochastic mirror descent algorithm applied with a non-smooth mirror map.

Within the context of sparse training, we also highlight the relevance of genetic evolutionary algorithms, particularly Sparse Evolutionary Training (SET) (Mocanu et al., 2018). SET applies a dynamic sparse training approach by performing pruning at the end of each epochremoving a fraction of the active connectionsand regrowing an equal number of new connections at random positions. This iterative process maintains a fixed sparsity level. Beyond sparse training, evolutionary algorithms have also been successfully applied to Neural Architecture Search (Miikkulainen et al., 2024; Elsken et al., 2019); however, they typically lack rigorous convergence guarantees.

In this paper, we propose a training algorithm based on linearized Bregman iterations, designed to promote sparsity in neural networks *during training*. A key benefit of Bregman iterations over regularization methods like the Lasso is that for the former the number of non-zero parameters of the trained networks usually increases monotonically. Hence, algorithms like LinBreg lend themselves to exploiting sparsity early on in the training process. In our method, we periodically freeze the networks structure: that is, we restrict updates to parameters that are non-zero in the current iteration. This approach offers two key benefits. First, stricter sparsity is enforced than through LinBreg alone, as the number of non-zero parameters cannot increase during the frozen phases. Second, during the frozen phases only derivatives corresponding to active parameters are required which provides scope for significant computational savings during training.

We embed the resulting algorithm within a multilevel optimization framework (Nash, 2000), which enables us to leverage existing convergence theory. In particular, we adapt the convergence analysis of Multilevel Bregman Proximal Gradient Descent (Elshiaty & Petra, 2026) to our sparse training setup. We find that our method is at least as performative as the standard LinBreg algorithm, while primarily yielding even sparser models at matched accuracy.

The remainder of this paper is structured as follows. First, we explain our algorithm and present how it can be interpreted as a multilevel optimization method. We proceed by proving a convergence result for the algorithm. Finally, we perform numerical experiments comparing our method to other methods that aim at achieving sparse but performative models. We make the following contributions:

- We propose a sparse training algorithm based on linearized Bregman iterations that periodically freezes the network structure during optimization.

- We interpret the resulting method as a multilevel stochastic optimization scheme and study convergence of the loss under suitable assumptions on the loss and the gradient estimators.

- We evaluate the proposed algorithm on benchmark image classification tasks and show that it produces sparse models with competitive performance. We further demonstrate reduced CPU training time when using sparsity-aware implementations.

- We show that the proposed multilevel strategy consistently performs at least as well as the standard LinBreg algorithm, while in most cases yielding sparser models and maintaining or improving accuracy.

## 2 Related Work

**Bregman Iterations / Mirror descent**  Bregman iterations were originally introduced by Osher et al. (2005) as iterative reconstruction method for imaging inverse problems to overcome the bias of regularization methods like total variation denoising (Rudin et al., 1992). Later they were applied to compressed sensing (Yin et al., 2008) and nonlinear inverse problems (Bachmayr & Burger, 2009; Benning et al., 2021). In the context of machine learning, they were used for sparsity of neural network representations (Bungert et al., 2021; 2022; Heeringa et al., 2025b;a) as well as for training networks with non-smooth activations of proximal

type (Wang & Benning, 2023). While Bregman iterations in their original form generalize the implicit Euler method, so-called linearized Bregman iterations are closely related to mirror descent (Nemirovskij & Yudin, 1983) or more precisely to lazy mirror descent / Nestorov's dual averaging (Nesterov, 2009). The method is also referred to as Bregman proximal gradient descent and a stochastic gradient version of it was coined LinBreg by Bungert et al. (2022). It must be emphasized that, just like different communities use different terminologies, they also developed different mathematical tools to analyze the convergence behavior. In particular, the inverse problems community put a lot of effort into analyzing Bregman iterations with sparsity-promoting regularizers which translates to mirror descent with non-smooth mirror maps. This will also be our approach in this paper.

**Multilevel Optimization**   Multilevel optimization methods originate from multigrid techniques, which were initially developed to solve differential equations efficiently. The MGOPT algorithm (Nash, 2000) was among the first to adapt these ideas to optimization problems. More recently, Elshiaty & Petra (2026) extended this framework by incorporating linearized Bregman iterations. Their work provides convergence guarantees via a PolyakŁojasiewicz-type inequality for the ML BPGD algorithm and demonstrates its effectiveness in image reconstruction tasks. Hovhannisyan et al. (2016) provide a connection between multilevel optimization and mirror descent, noting that the latter is equivalent to linearized Bregman iterations. They further incorporate acceleration techniques, prove convergence of their algorithm, and illustrate its performance through numerical experiments on face recognition. Multilevel strategies have also been explored in deep learning, particularly for training residual neural networks (ResNets). Kopaniáková & Krause (2023) introduce a hierarchy based on networks of varying depth and width, leveraging the fact that smaller models are cheaper to train. Similar approaches have been made, e.g., by Gaedke-Merzhäuser et al. (2021). Other approaches embed the multilevel hierarchy into the objective function rather than the model architecture. For example, Braglia et al. (2020) use varying batch sizes to compute the loss, effectively inducing a multiscale structure during training.

**Sparse neural networks**   Within the context of sparse training, several methods have been proposed to obtain sparse yet performant models without the need to train a dense network beforehand. One approach is to keep a static sparsity pattern throughout training which is identified in advance based upon, say, the magnitude of the initialized weights' gradient of the loss (Lee et al., 2019) or second order derivative information (Singh & Alistarh, 2020). Another way is to allow for the sparsity pattern to dynamically change throughout training by including mechanisms that remove and regrow connections according to certain criterion. For example, DEEP-R (Bellec et al., 2018) uses a random regrowth strategy whereas RigL (Evci et al., 2020) activates connections according to gradient magnitude. Other criterion may include: accumulated gradient information (Liu et al., 2021); momentum (Dettmers & Zettlemoyer, 2019); the difference to the weights of a previous training step (Li et al., 2023). DFBST (Pote et al., 2023) applies binary masks during both the forward and backward passes in which the forward pass mask sparsifies the weights, while the backward pass mask restricts gradient updates, enabling sparse training. Top-KAST (Jayakumar et al., 2020) maintains a fixed sparse model by using only the top-$D$ weights for the forward pass, but employs a slightly larger set of parameters during the backward pass to update and explore potential alternative masks. Ji et al. (2024b) decouple the optimization of masks and weights, treating the mask as an explicit optimization variable. Such a decoupling can be viewed in a time-dependent mirror flow framework (Jacobs & Burkholz, 2025) in which convergence and optimality results can be stated. The AC/DC method (Peste et al., 2021) uses alternating phases of full and sparse support optimization to produce accurate sparsedense model pairs, while providing convergence guarantees of the underlying iterative hard thresholding algorithm for non-convex training under a Polyak–Łojasiewicz-type inequality. Yin et al. (2023) propose to obtain channel-level sparse networks by performing channel pruning during training, where channels are removed based on the mean magnitude of their weights. Techniques that further improve existing sparse training algorithms include the application of the soft thresholding operator with learnable thresholds to layer weights (Kusupati et al., 2020) and the performance of weight updates that minimize in a local neighborhood (Ji et al., 2024a).

## 3 Method

A typical training problem to find optimal network parameters $\theta \in \mathbb{R}^d$ consists of solving

$$\min_{\theta \in \mathbb{R}^d} \mathcal{L}(\theta), \tag{1}$$

where $\mathcal{L} \colon \mathbb{R}^d \to \mathbb{R}$ is a differentiable and non-negative loss function, for example, the empirical loss. One way to enforce constraints or encourage sparsity in solutions is to specify a proper, convex, and lower semicontinuous function $J \colon \mathbb{R}^d \to (-\infty, \infty]$, such as the $\ell_1$-norm or indicator function of a closed convex set, and consider a minimization of $\mathcal{L} + J$. An alternative approach, which we discuss in the following, is to employ (Linearized) Bregman iterations, in which the loss $\mathcal{L}$ is directly minimized while implicitly minimizing $J$.

### 3.1 Linearized Bregman iterations

Solving (1) with Bregman iterations requires iterating

$$
\begin{aligned}
\theta^{(k+1)} &= \arg\min_{\theta \in \mathbb{R}^d} D_J^{p^{(k)}}(\theta, \theta^{(k)}) + \tau^{(k)}\mathcal{L}(\theta), \\
p^{(k+1)} &= p^{(k)} - \tau^{(k)}\nabla\mathcal{L}(\theta^{(k+1)}) \in \partial J(\theta^{(k+1)}),
\end{aligned}
\tag{2}
$$

where the so-called Bregman divergence (associated to $J$) is defined as:

$$D_J^p(\tilde{\theta}, \theta) := J(\tilde{\theta}) - J(\theta) - \langle p, \tilde{\theta} - \theta \rangle. \tag{3}$$

Here $\theta \in \mathrm{dom}(\partial J)$, $p \in \partial J(\theta)$ is a subgradient, and $\tau^{(k)} > 0$ is a sequence of step sizes. The Bregman divergence (3) can be interpreted as the difference between $J$ and its linearization around $\theta$ and satisfies properties such as $D_J^p(\theta, \theta) = 0$ and, due to convexity of $J$, $D_J^p(\tilde{\theta}, \theta) \geq 0$ as long as $p$ is a subgradient.

Since the optimization problem (2) is almost as hard to solve as (1), one can replace $\mathcal{L}$ in (2) with its first order approximation $\mathcal{L}(\theta^{(k)}) + \langle \nabla\mathcal{L}(\theta^{(k)}), \theta - \theta^{(k)} \rangle$ and $J$ with the strongly convex elastic-net regularizer

$$J_\delta(\theta) := \frac{1}{2\delta}\|\theta\|^2 + J(\theta), \quad \delta \in (0, \infty).$$

Doing so yields Linearized Bregman iterations

$$v^{(k+1)} = v^{(k)} - \tau\nabla\mathcal{L}(\theta^{(k)}), \tag{4a}$$

$$\theta^{(k+1)} = \mathrm{prox}_{\delta J}(\delta v^{(k+1)}), \tag{4b}$$

where $v^{(0)} \in \partial J_\delta(\theta^{(0)})$ for some initial point $\theta^{(0)}$ and

$$\mathrm{prox}_{\delta J}(\theta) := \arg\min_{\tilde{\theta} \in \mathbb{R}^d} \frac{1}{2\delta}\|\tilde{\theta} - \theta\|^2 + J(\tilde{\theta}) \tag{5}$$

is the proximal operator. To make (4) feasible for the high-dimensional and non-convex problems arising in machine learning, Bungert et al. (2022) in their LinBreg method replace the gradient $\nabla\mathcal{L}(\theta^{(k)})$ in (4a) with an unbiased stochastic estimator and provide convergence analysis.

Note that if $J \equiv 0$, the proximal operator is the identity map and, additionally taking $\delta = 1$, (4) recovers Gradient Descent. More generally, (4) coincides with mirror descent (Beck & Teboulle, 2003) applied to the distance generating function $J_\delta$. To see this, notice that $\nabla J_\delta^* = \mathrm{prox}_{\delta J}(\delta \cdot)$, where $J_\delta^*$ denotes the convex conjugate (Bauschke & Combettes, 2011) of $J_\delta$.

Although evaluating the proximal operator in (4b) is phrased as a minimization problem, particular choices of $J$ admit closed forms, e.g., $J = \lambda\|\cdot\|_1$ yields the soft shrinkage operator

$$\mathrm{prox}_{\delta J}(\delta v) = \delta \, \mathrm{sign}(v) \max(|v| - \lambda, 0) \tag{6}$$

applied componentwise. This particular choice also demonstrates how Bregman iterations can lead to sparse networks. In (6), only parameters whose corresponding dual variable $v$ exceeds the threshold $\lambda$ in absolute value will be non-zero. Thus, we can view (4b) as a pruning step inherent to the optimizer, where the pruning criterion employs information associated with the regularizer $J$, and not just the magnitude of the parameter or of the gradient of the training loss $\mathcal{L}$.

In practice, $\theta$ can represent the parameters of several network layers, and as such one may wish to employ a different regularizer for each layer. To represent this, we split the parameter vector $\theta$ into $G$ groups via $\theta = (\theta_{(1)}, \ldots, \theta_{(G)})$, where each group $\theta_{(g)} \in \mathbb{R}^{d_g}$ contains $d_g$ scalar parameters. Furthermore, we assume the regularizer $J$ acts on these groups separately, taking the form

$$J(\theta) = \sum_{g=1}^{G} J_g(\theta_{(g)}), \tag{7}$$

where each $J_g \colon \mathbb{R}^{d_g} \to (-\infty, \infty]$ is proper, convex, and lower-semicontinuous. We see that (7) includes the standard $\ell_1$-norm but also the group $\ell_{1,2}$-norm (Scardapane et al., 2017), given by

$$J(\theta) := \sum_{g=1}^{G} \sqrt{n_g} \|\theta_{(g)}\|_2, \tag{8}$$

where $n_g$ denotes the number of parameters in the group. While the $\ell_1$-norm encourages individual parameter sparsity, the group $\ell_{1,2}$-norm can be used to force entire structures, e.g., convolutional kernels, to vanish.

### 3.2 A Multilevel Approach

The main idea of our algorithm is to exploit the induced sparsity of the iterations (4) by only performing a full update with this rule every $m$ iterations and otherwise only updating the non-zero parameters. Consequently, most training steps of our proposed method will only require gradients with respect to the active parameters which, depending on the induced sparsity pattern, provides scope for significant computational savings during training. We show that we can interpret this idea as a multilevel optimization scheme, allowing us to adapt convergence proofs from the multilevel optimization literature.

More precisely, we consider a two-level framework consisting of the actual minimization problem (1) and a coarse problem with fewer variables. To map between these levels, restriction and prolongation operators are used. The restriction operator at iteration $k$, denoted $R^{(k)} \colon \mathbb{R}^d \to \mathbb{R}^{D_k}$ with $D_k < d$, is a linear map that selects a subset of parameters for optimization. Consequently, the rows of the matrix $R^{(k)}$ are standard unit vectors and $\theta_i$ is selected by $R^{(k)}$ if and only if one of the rows of the matrix is the $i$-th standard unit vector. We consider the case where entire groups are selected by the restriction operator, so if one component of a group $\theta_{(g)}$ is selected, then all the other components must be selected as well. For example, if any parameter of a single convolutional kernel is non-zero, then we allow for the entire kernel to be updated - not just the non-zero elements. To formalize this, given the number of selected groups $G^{(k)}$, we can define an injective function $r^{(k)} \colon \{1, 2, \ldots, G^{(k)}\} \to \{1, 2, \ldots, G\}$ such that

$$R^{(k)}\theta = (\theta_{(r^{(k)}(1))}, \ldots, \theta_{(r^{(k)}(G^{(k)}))}), \quad \theta \in \mathbb{R}^d.$$

For a given coarse variable $\hat{\theta}$, this function determines where its parameter groups belong on the fine level.

The corresponding prolongation operator $P^{(k)} \colon \mathbb{R}^{D_k} \to \mathbb{R}^d$ maps from the coarse to the fine level and is defined as the transpose of the restriction $P^{(k)} = (R^{(k)})^T$. This means that the prolongation operator maps the groups that were selected by the restriction back and simply completes the parameter vector $\theta$ by setting zero for every parameter group that was not selected by the restriction. Using the previously defined function $r^{(k)}$, we can explicitly write the prolongation of a coarse variable $\hat{\theta} \in \mathbb{R}^{D_k}$ as

$$(P^{(k)}\hat{\theta})_{(g)} = \begin{cases} \hat{\theta}_{(i)}, & \text{if } g = r^{(k)}(i), \\ 0, & \text{otherwise.} \end{cases}$$

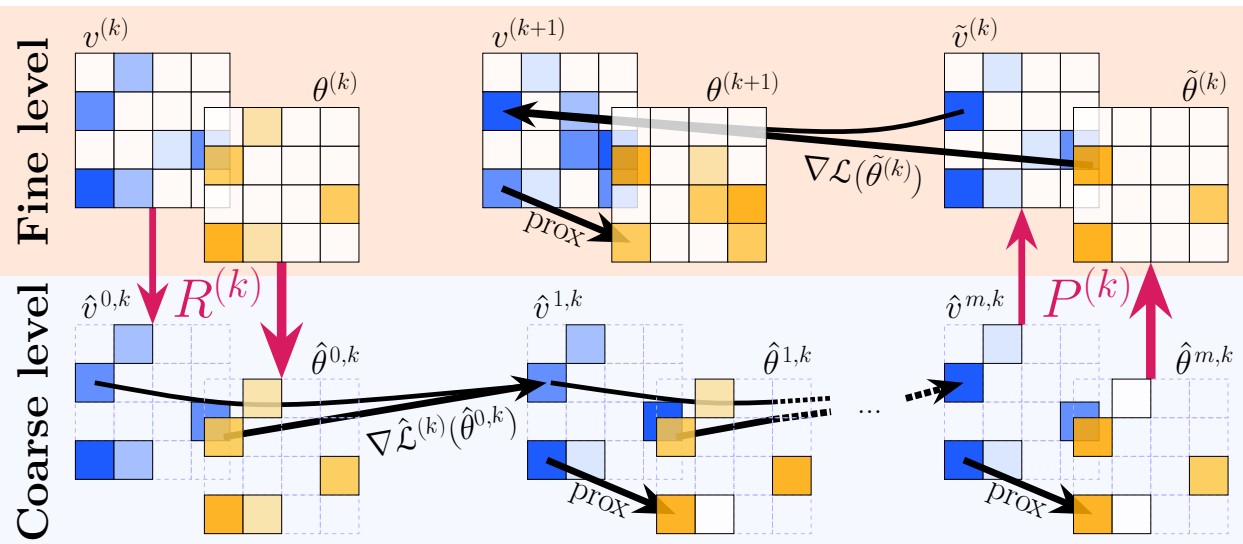

Figure 1: Visualization of the Multilevel LinBreg algorithm. Only parameters selected by the restriction operator $R^{(k)}$ will be updated by LinBreg for $m$ iterations on the coarse level. Then, we use the prolongation operator $P^{(k)}$ to cast the updated restricted parameters back to the fine level, i.e., the full parameter space, after which we perform one LinBreg update.

While the case where the restriction operator chooses the groups that are non-zero at iteration $k$ is the most interesting for us, our analysis works for arbitrary selections of parameter groups.

In our algorithm, during each iteration $k$, the restriction operator $R^{(k)}$ maps the current iterate $\theta^{(k)}$ and the corresponding subgradient $v^{(k)}$ to the coarse level. We denote these restrictions by $\hat{\theta}^{0,k} := R^{(k)}\theta^{(k)}$ and $\hat{v}^{0,k} := R^{(k)}v^{(k)}$. The algorithm then performs $m$ LinBreg steps to minimize the coarse loss

$$\hat{\mathcal{L}}^{(k)}(\hat{\theta}) := \mathcal{L}(\theta^{(k)} + P^{(k)}(\hat{\theta} - \hat{\theta}^{0,k})) \tag{9}$$

using $\hat{J}^{(k)}_\delta(\hat{\theta}) := \frac{1}{2\delta}\|\hat{\theta}\|^2 + \hat{J}^{(k)}(\hat{\theta})$ as the regularizer, where

$$\hat{J}^{(k)}(\hat{\theta}) := \sum_{i=1}^{G^{(k)}} J_{r^{(k)}(i)}(\hat{\theta}_{(i)}). \tag{10}$$

The result of these coarse iteration steps $\hat{\theta}^{m,k}$ with corresponding subgradient $\hat{v}^{m,k}$ is then mapped back to the fine level via

$$\tilde{\theta}^{(k+1)} := \theta^{(k)} + P^{(k)}(\hat{\theta}^{m,k} - \hat{\theta}^{0,k}), \qquad \tilde{v}^{(k+1)} := v^{(k)} + P^{(k)}(\hat{v}^{m,k} - \hat{v}^{0,k}), \tag{11}$$

before performing a LinBreg step on the fine level to obtain $\theta^{(k+1)}$ and $v^{(k+1)}$. The full Multilevel LinBreg algorithm is presented in Algorithm 1 and visualized in Figure 1.

If the restriction operator at iteration $k$ only selects the parameter groups that are non-zero for $\theta^{(k)}$, the coarse loss (9) and the prolongation to the fine level simplify to

$$\hat{\mathcal{L}}^{(k)}(\hat{\theta}) = \mathcal{L}(P^{(k)}\hat{\theta}), \qquad \tilde{\theta}^{(k+1)} = P^{(k)}\hat{\theta}^{m,k}.$$

For the subgradients, however, such simplification is not available.

Typically in multilevel optimization, a linear correction term is added to the coarse objective function to ensure first-order coherence, i.e., critical points on the fine level are transferred to ones on the coarse level. In our case, due to the special structure of $\hat{\mathcal{L}}^{(k)}$ no correction term is necessary, since

$$\nabla\hat{\mathcal{L}}^{(k)}(\hat{\theta}^{0,k}) = R^{(k)}\nabla\mathcal{L}(\theta^{(k)}).$$

---

**Algorithm 1** Multilevel LinBreg

---

**Input:** Initial guess $\theta^{(0)} \in \mathbb{R}^d, v^{(0)} \in \partial J_\delta(\theta^{(0)})$
**for** $k \in \mathbb{N}$ **do**
$\quad \hat{\theta}^{0,k} = R^{(k)}\theta^{(k)}$
$\quad \hat{v}^{0,k} = R^{(k)}v^{(k)}$
$\quad$ **for** $i = 1, \ldots, m$ **do**
$\quad\quad \hat{g}^{(i,k)} \leftarrow$ estimator of $\nabla\hat{\mathcal{L}}^{(k)}(\hat{\theta}^{i-1,k})$
$\quad\quad \hat{v}^{i,k} = \hat{v}^{i-1,k} - \hat{\tau}^{i-1,k}\hat{g}^{(k)}$
$\quad\quad \hat{\theta}^{i,k} = \text{prox}_{\delta\hat{j}^{(k)}}(\delta\hat{v}^{i,k})$
$\quad$ **end for**
$\quad \tilde{v}^{(k+1)} = v^{(k)} + P^{(k)}(\hat{v}^{m,k} - \hat{v}^{0,k})$
$\quad \tilde{\theta}^{(k+1)} = \theta^{(k)} + P^{(k)}(\hat{\theta}^{m,k} - \hat{\theta}^{0,k})$
$\quad g^{(k+1)} \leftarrow$ estimator of $\nabla\mathcal{L}(\tilde{\theta}^{(k+1)})$
$\quad v^{(k+1)} = \tilde{v}^{(k+1)} - \tau g^{(k+1)}$
$\quad \theta^{(k+1)} = \text{prox}_{\delta J}(\delta v^{(k+1)})$
**end for**

---

Hence, if $\theta^{(k)}$ is a critical point of $\mathcal{L}$, then $\hat{\theta}^{0,k} := R^{(k)}\theta^{(k)}$ is a critical point of $\hat{\mathcal{L}}^{(k)}$.

Typically, in multilevel optimization, a criterion is used to determine whether the coarse model should be employed, with the aim of avoiding coarse updates when they are not expected to be beneficial. Such criteria are usually based on the norm of the restricted gradient; see, e.g., Wen & Goldfarb (2009); Vanmaele et al. (2025); Hovhannisyan et al. (2016). However, for simplicity, we do not use such a criterion, but instead perform $m$ coarse-level steps after every fine-level iteration.

Since we work with subgradients, we carefully investigate the transfer between the different levels. In particular, we show that restricting the subgradients from the fine level yields subgradients of the coarse regularizer and mapping the coarse subgradients back to the fine level as described also leads to subgradients on the fine level due to the structure of $J$ from (7). The proofs of the following propositions can be found in Appendix A.1.

**Proposition 3.1.** *If $v^{(k)} \in \partial J_\delta(\theta^{(k)})$, then defining $\hat{v}^{0,k} := R^{(k)}v^{(k)}$ and $\hat{\theta}^{0,k} := R^{(k)}\theta^{(k)}$ we have $\hat{v}^{0,k} \in \partial\hat{J}_\delta^{(k)}(\hat{\theta}^{0,k})$.*

This implies that in Algorithm 1 we start the coarse iteration with a feasible pair $(\hat{\theta}^{0,k}, \hat{v}^{0,k})$. Consequently, due to the structure of the iteration, we subsequently obtain $\hat{v}^{i,k} \in \partial\hat{J}_\delta^{(k)}(\hat{\theta}^{i,k})$ for $i = 1, \ldots, m$. We also observe that the way of transferring the updated variables back to the fine level preserves the fact that $\tilde{v}^{(k+1)}$ is a subgradient of $J_\delta$ at $\tilde{\theta}^{(k+1)}$.

**Proposition 3.2.** *If $\hat{\theta}^{m,k} \in \text{dom}(\partial\hat{J}^{(k)})$ and $\hat{v}^{m,k} \in \partial\hat{J}_\delta^{(k)}$, then defining*

$$\tilde{\theta}^{(k+1)} := \theta^{(k)} + P^{(k)}(\hat{\theta}^{m,k} - \hat{\theta}^{0,k}),$$
$$\tilde{v}^{(k+1)} := v^{(k)} + P^{(k)}(\hat{v}^{m,k} - \hat{v}^{0,k}),$$

*we have $\tilde{v}^{(k+1)} \in \partial J_\delta(\tilde{\theta}^{(k+1)})$.*

To conclude this section, we would like to highlight the key differences between our algorithm and the methods ML BPGD (Elshiaty & Petra, 2026), MGOPT (Nash, 2000), and AC/DC (Peste et al., 2021), starting with the former two. Firstly, we use adaptive restriction and prolongation operators, which are necessary to focus on different parameters at different iterations. Secondly, we employ stochastic gradient estimators instead of exact gradients. Thirdly, we do not perform a line search when mapping from the coarse level to the fine level. MGOPT and ML BPGD both use arbitrary convex coarse objective functions with a linear correction term to ensure that the direction found with the coarse iterations is a descent direction, as well as a line search to ensure that the fine loss actually decreases. Our specific choice of the coarse objective eliminates the need for a line search. Finally, unlike ML BPGD, our regularizer $J_\delta$ can be non-differentiable. Therefore,

rather than working with gradients, we need to consider subgradients which must be handled with care when mapping between different levels (see Propositions 3.1 and 3.2).

AC/DC by Peste et al. (2021) is a sparse training algorithm that alternates between dense and sparse phases similar to our proposed method. In fact, with some minor changes, our method recovers a version of AC/DC. To be precise, changing the transfer to the fine level (11) and the coarse objective (9) to

$$\tilde{\theta}^{(k+1)} = P^{(k)}\hat{\theta}^{m,k}, \quad \tilde{v}^{(k+1)} = P^{(k)}\hat{v}^{m,k}, \quad \hat{\mathcal{L}}^{(k)}(\hat{\theta}) = \mathcal{L}(P^{(k)}\hat{\theta}),$$

and using SGD as the optimizer (i.e., $J = 0$ and $\delta = 1$) yields an AC/DC type algorithm, where the dense phases only consist of one step and the compressed phase corresponds to the $m$ steps on the coarse level. This means that a key difference between the two algorithms is that our method retains previous information about components that are not selected in the sparse phase, whereas AC/DC discards said information entirely. Additionally, the restriction operator that we use in practice is, depending on the regularizer, based upon a soft-thresholding type operator whereas the restriction operator in AC/DC is based upon hard thresholding.

## 4 Convergence Analysis

For the convergence analysis, we need to make further assumptions. We follow Bauschke et al. (2019); Elshiaty & Petra (2026) and require the loss function to be smooth relative to the regularizer and to satisfy a Polyak–Łojasiewicz-type inequality.

**Assumption 1.** *We assume that $\mathcal{L}$ is L-smooth with respect to $J_\delta$, i.e.*

$$\mathcal{L}(\tilde{\theta}) \leq \mathcal{L}(\theta) + \langle \nabla\mathcal{L}(\theta), \tilde{\theta} - \theta \rangle + LD_{J_\delta}^v(\tilde{\theta}, \theta)$$

*for $\tilde{\theta} \in \mathbb{R}^d$, $\theta \in \mathrm{dom}(\partial J)$ and $v \in \partial J_\delta(\theta)$.*

Since $J_\delta$ is strongly convex, its induced Bregman divergence is bounded from below by the squared Euclidean norm. Therefore, from the descent lemma for $L$-smooth functions (Bauschke & Combettes, 2011; Beck, 2017) it follows that this assumption holds in particular for loss functions $\mathcal{L}$ with a Lipschitz-continuous gradient which, however, is generally a much stronger condition than Assumption 1.

**Assumption 2.** *For $\theta \in \mathrm{dom}(\partial J), v \in \partial J_\delta(\theta)$ and $\tau > 0$, let $v_\tau^+ := v - \tau\nabla\mathcal{L}(\theta)$ and $\theta_\tau^+ := \mathrm{prox}_{\delta J}(\delta v_\tau^+)$. We assume the existence of a function $\lambda\colon (0,\infty) \to (0,\infty)$ and some $\eta > 0$ such that*

$$D_{J_\delta}^{v_\tau^+}(\theta, \theta_\tau^+) \geq \lambda(\tau)D_{J_\delta}^{v_1^+}(\theta, \theta_1^+) \tag{12}$$

*for $\theta \in \mathrm{dom}(\partial J), v \in \partial J_\delta(\theta), \tau > 0$ and*

$$D_{J_\delta}^{v_1^+}(\theta, \theta_1^+) \geq \eta(\mathcal{L}(\theta) - \mathcal{L}^\star), \tag{13}$$

*where $\mathcal{L}^\star := \inf_{\mathbb{R}^d} \mathcal{L}$.*

In the gradient descent case where $J_\delta = \frac{1}{2}\|\cdot\|^2$, assumption (12) is satisfied with $\lambda(\tau) = \tau^2$ and (13) becomes the well-known Polyak–Łojasiewicz inequality $\|\nabla\mathcal{L}(\theta)\|^2 \geq 2\eta(\mathcal{L}(\theta) - \mathcal{L}^\star)$, which is weaker than convexity but requires every stationary point of $\mathcal{L}$ to be a minimizer. Assumption 2 has been been employed in Bauschke et al. (2019) and Elshiaty & Petra (2026). Moreover, Bauschke et al. (2019) discuss different classes of functions for which it is satisfied. In particular, they show that relative strong convexity with respect to $J_\delta$ is sufficient for (13). The concept of relative strong convexity with respect to the regularizer $J_\delta$ was also employed by Bungert et al. (2022) in their convergence analysis of the LinBreg algorithm and they showed in their Remark 5 that relative strong convexity is locally implied by standard strong convexity around a minimizer in the case of sparsity regularization, i.e., $J(\theta) = \|\theta\|_1$.

For our convergence analysis, we assume that the gradient estimators have a bounded mean squared error. To be precise, let $(\Omega, \mathcal{F}, \mathbb{P})$ be a probability space and $g : \mathbb{R}^d \times \Omega \to \mathbb{R}^d, \hat{g}^{(k)} : \mathbb{R}^{D_k} \times \Omega \to \mathbb{R}^{D_k}$. We make the following assumption.

**Assumption 3.** *We assume that there exist constants $\sigma_f, \sigma_c \geq 0$ such that*

$$\mathbb{E}\left[\|g(\theta, \omega) - \nabla\mathcal{L}(\theta)\|^2 \,|\, \theta\right] \leq \sigma_f^2, \qquad \mathbb{E}\left[\|\hat{g}^{(k)}(\hat{\theta}, \omega) - \nabla\hat{\mathcal{L}}^{(k)}(\hat{\theta})\|^2 \,|\, \hat{\theta}\right] \leq \sigma_c^2$$

*for any $k \in \mathbb{N}$ and $\theta \in \mathbb{R}^d, \hat{\theta} \in \mathbb{R}^{D_k}$.*

Additionally, we define filtrations

$$\mathcal{F}_k := \sigma\left(\theta^{(0)}, \{\hat{\theta}^{i,j}\}_{i=0,\ldots,m}^{j<k}, \{\tilde{\theta}^{(j)}, \theta^{(j)}\}_{j=1}^k\right),$$

$$\mathcal{G}_k := \sigma(\mathcal{F}_k, \theta^{0,k}, \ldots, \theta^{m,k}, \tilde{\theta}^{(k+1)}),$$

such that $\mathcal{F}_k$ contains all randomness until iteration $k$ and $\mathcal{G}_k$ additionally contains the randomness of the coarse iteration $k \to k+1$. These filtrations in particular satisfy $\mathcal{F}_k \subset \mathcal{G}_k \subset \mathcal{F}_{k+1}$. Furthermore, we assume that whenever a gradient estimator is evaluated, it is computed using a fresh sample drawn independently of the past. Specifically, $g^{(k+1)} := g(\tilde{\theta}^{(k+1)}, \omega_{k+1})$ where $\omega_{k+1} \sim \mathbb{P}$ is independent of $\mathcal{G}_k$. Likewise, for the step $i$ on the coarse level in iteration $k$, $\hat{g}^{(i,k)} := \hat{g}^{(k)}(\hat{\theta}^{i,k}, \xi_{i,k})$ where $\xi_{i,k} \sim \mathbb{P}$ is independent of $\mathcal{F}^{i,k} := \sigma(\mathcal{F}_k, \hat{\theta}^{0,k}, \ldots, \hat{\theta}^{i,k})$.

For the proof of our main result Theorem 4.4 we require two more lemmas, the proofs of which are very similar to the corresponding results in Bauschke et al. (2019); Elshiaty & Petra (2026), with the key differences being that we utilize subgradients instead of classical gradients of the regularizer $J_\delta$, and that we use stochastic gradients. In particular, unlike Elshiaty & Petra (2026), we need to investigate the transfer of subgradients between the levels and prove that our coarse objective inherits relative smoothness from the $\mathcal{L}$.

Before carrying out our convergence analysis, we introduce the concept of the symmetrized Bregman divergence by summing up two Bregman divergences,

$$D_{J_\delta}^{\mathrm{sym}}(\theta, \tilde{\theta}) := D_{J_\delta}^v(\tilde{\theta}, \theta) + D_{J_\delta}^{\tilde{v}}(\theta, \tilde{\theta}) = \langle v - \tilde{v}, \theta - \tilde{\theta}\rangle, \tag{14}$$

for $\theta, \tilde{\theta} \in \mathrm{dom}\partial J$ and $v \in \partial J_\delta(\theta), \tilde{v} \in \partial J_\delta(\tilde{\theta})$, where we suppress the dependency on the subgradients in the notation.

First, we investigate the loss during the fine level updates.

**Lemma 4.1.** *Let Assumptions 1, 2 and 3 be satisfied. Then, if $0 < \tau < \frac{1}{2L}$,*

$$\mathbb{E}\left[\mathcal{L}(\theta^{(k+1)}) - \mathcal{L}^\star \,|\, \mathcal{G}_k\right] \leq \left(1 - \frac{\eta\lambda(\tau)}{\tau}\right)(\mathcal{L}(\tilde{\theta}^{(k+1)}) - \mathcal{L}^\star) + \tau\delta\sigma_f^2.$$

*Proof.* Using Assumption 1, Young's inequality, and Proposition 3.2, we can estimate

$$\mathcal{L}(\theta^{(k+1)}) - \mathcal{L}(\tilde{\theta}^{(k+1)})$$

$$\leq \langle g^{(k+1)}, \theta^{(k+1)} - \tilde{\theta}^{(k+1)}\rangle + \langle \nabla\mathcal{L}(\tilde{\theta}^{(k+1)}) - g^{(k+1)}, \theta^{(k+1)} - \tilde{\theta}^{(k+1)}\rangle + LD_{J_\delta}^{\tilde{v}^{(k+1)}}(\theta^{(k+1)}, \tilde{\theta}^{(k+1)})$$

$$\leq -\frac{1}{\tau}D_{J_\delta}^{sym}(\theta^{(k+1)}, \tilde{\theta}^{(k+1)}) + \delta\tau\|\nabla\mathcal{L}(\tilde{\theta}^{(k+1)}) - g^{(k+1)}\|^2 + \left(L + \frac{1}{2\tau}\right)D_{J_\delta}^{\tilde{v}^{(k+1)}}(\theta^{(k+1)}, \tilde{\theta}^{(k+1)}).$$

Taking the conditional expectation with respect to $\mathcal{G}_k$ and combining one part of the symmetric Bregman distance with the last term on the right yields

$$\mathbb{E}\left[\mathcal{L}(\theta^{(k+1)}) - \mathcal{L}(\tilde{\theta}^{(k+1)}) \,|\, \mathcal{G}_k\right] \leq -\frac{1}{\tau}\mathbb{E}\left[D_{J_\delta}^{v^{(k+1)}}(\tilde{\theta}^{(k+1)}, \theta^{(k+1)}) \,|\, \mathcal{G}_k\right]$$

$$-\frac{1 - 2L\tau}{2\tau}\mathbb{E}\left[D_{J_\delta}^{\tilde{v}^{(k+1)}}(\theta^{(k+1)}, \tilde{\theta}^{(k+1)}) \,|\, \mathcal{G}_k\right] + \delta\tau\sigma_f^2$$

$$\leq -\frac{\eta\lambda(\tau)}{\tau}\left(\mathcal{L}(\tilde{\theta}^{(k+1)}) - \mathcal{L}^\star\right) + \delta\tau\sigma_f^2,$$

where we used Assumptions 2 and 3 together with the step-size bound. Subtracting $\mathcal{L}^\star$ from both sides gives the desired result. $\square$

The second lemma investigates the decay of the function value from a coarse update. The key observation is that our coarse objective $\hat{\mathcal{L}}^{(k)}$ is $L$-smooth relative to the coarse regularizer $\hat{J}_\delta^{(k)}$, which follows from Assumption 1 and Lemma A.1. The proof of the following lemma is deferred to Appendix A.2.

**Lemma 4.2.** *Let Assumption 1 and Assumption 3 be satisfied. If the step sizes are chosen such that $\hat{\tau}^{i,k} \leq \frac{1}{4L}$, then the coarse updates in iteration $k$ yield*

$$\mathbb{E}\left[\mathcal{L}(\tilde{\theta}^{(k+1)}) \,|\, \mathcal{F}_k\right] \leq \mathcal{L}(\theta^{(k)}) - \sum_{j=0}^{m-1} \frac{1}{4\hat{\tau}^{j,k}} \mathbb{E}\left[D_{\hat{J}_\delta^{(k)}}^{\mathrm{sym}}(\hat{\theta}^{j+1,k}, \hat{\theta}^{j,k}) \,|\, \mathcal{F}_k\right] + \frac{\delta\sigma_c^2}{2} \sum_{j=0}^{m-1} \hat{\tau}^{j,k}.$$

*Remark* 4.3. In the deterministic case ($\sigma_c = 0$), the previous result guarantees a decrease in the loss during the coarse updates. The resulting estimate closely parallels the loss decay established in (Elshiaty & Petra, 2026).

**Theorem 4.4.** *Let Assumptions 1, 2 and 3 be satisfied and $(\theta^{(k)})$ be the sequence generated by Algorithm 1 If the fine step size $\tau$ is chosen such that $\tau < \frac{1}{2L}$ and the coarse step sizes such that $\hat{\tau}^{i,k} \leq \frac{1}{4L}$ then*

$$\mathbb{E}\left[\mathcal{L}(\theta^{(k)}) - \mathcal{L}^\star\right] \leq (1-r)^k(\mathcal{L}(\theta^{(0)}) - \mathcal{L}^\star) + \sum_{i=0}^{k-1}(1-r)^{k-i}\hat{\rho}_i + \frac{\tau^2\delta\sigma_f^2}{\eta\lambda(\tau)},$$

*where $r = \frac{\eta\lambda(\tau)}{\tau}$ and*

$$\hat{\rho}_i = \sum_{j=0}^{m-1} \left( \frac{\delta\sigma_c^2}{2}\hat{\tau}^{j,i} - \frac{1}{4\hat{\tau}^{j,i}}\mathbb{E}\left[D_{\hat{J}_\delta^{(k)}}^{\mathrm{sym}}(\hat{\theta}^{j+1,i}, \hat{\theta}^{j,i})\right] \right).$$

*Proof.* Combining Lemma 4.1, Lemma 4.2 and the tower property, we obtain

$$\mathbb{E}\left[\mathcal{L}(\theta^{(k+1)}) - \mathcal{L}^\star \,|\, \mathcal{F}_k\right] = \mathbb{E}\left[\mathbb{E}\left[\mathcal{L}(\theta^{(k+1)}) - \mathcal{L}^\star \,|\, \mathcal{G}_k\right] \,|\, \mathcal{F}_k\right]$$

$$\leq \left(1 - \frac{\eta\lambda(\tau)}{\tau}\right)\left(\mathbb{E}\left[\mathcal{L}(\tilde{\theta}^{(k+1)}) \,|\, \mathcal{F}_k\right] - \mathcal{L}^\star\right) + \tau\delta\sigma_f^2$$

$$\leq (1-r)\left(\mathcal{L}(\theta^{(k)}) - \mathcal{L}^\star\right) + (1-r)\sum_{j=0}^{m-1}\left(\frac{\delta\sigma_c^2}{2}\hat{\tau}^{j,k} - \frac{1}{4\hat{\tau}^{j,k}}\mathbb{E}\left[D_{\hat{J}_\delta^{(k)}}^{sym}(\hat{\theta}^{j+1,k}, \hat{\theta}^{j,k}) \,|\, \mathcal{F}_k\right]\right) + \tau\delta\sigma_f^2.$$

Iterating this inequality and utilizing the tower property gives the desired estimate. Notice that $\sum_{i=0}^{k}(1-r)^i \leq \sum_{i=0}^{\infty}(1-r)^i = \frac{\tau}{\eta\lambda(\tau)}$.

□

As a direct consequence, the objective values converge to the optimal value in expectation if we use exact gradients on the fine level and the coarse step-sizes converge to zero.

**Corollary 4.5.** *Let the assumptions from Theorem 4.4 hold and assume that we use exact gradients on the fine level (i.e. $\sigma_f = 0$). Define $\hat{\tau}^{(k)} := \max_{i=0,\ldots,m-1} \hat{\tau}^{i,k}$, the maximum stepsize on the coarse level in iteration $k$. If $\hat{\tau}^{(k)}$ converges to zero, then*

$$\lim_{k\to\infty} \mathbb{E}\left[\mathcal{L}(\theta^{(k)})\right] = \mathcal{L}^\star.$$

*Proof.* To prove this result, we consider the worst-case scenario in Theorem 4.4, by employing the estimate $\hat{\rho}_i \leq \frac{\delta\sigma_c^2}{2}\sum_{j=0}^{m-1}\hat{\tau}^{j,i}$ for any $i \in \mathbb{N}$. This is sharp if the potential benefit in the loss descent with the symmetrized Bregman divergence is zero. Hence,

$$\sum_{i=0}^{k-1}(1-r)^{k-i}\hat{\rho}_i \leq \sum_{i=0}^{k-1}(1-r)^{k-i}\sum_{j=0}^{m-1}\frac{\delta\sigma_c^2}{2}\hat{\tau}^{j,i} \leq C\sum_{i=0}^{k-1}(1-r)^{k-i}\hat{\tau}^{(i)}$$

$$= C\sum_{i=1}^{k}(1-r)^i\hat{\tau}^{(k-i)} = \sum_{i=1}^{\infty}\mathbb{1}_{i\leq k}(1-r)^i\hat{\tau}^{(k-i)}.$$

for some positive constant $C$ depending on $m, \delta$ and $\sigma_c^2$. To prove that the latter sum converges to zero as $k$ approaches infinity, we observe that for each $i \in \mathbb{N}$, the summands on the right hand side converge to zero as $k \to \infty$ and they are dominated by a constant times $(1-r)^i$ which is summable. By the dominated convergence theorem for series, the sum converges to zero. Applying Theorem 4.4 concludes the proof. $\square$

*Remark* 4.6. Without the assumption of using exact gradients for steps on the fine level, Theorem 4.4 does not allow to conclude that the expected gap between the achieved and optimal loss value converges to zero. However, under the assumptions of Corollary 4.5 with $\sigma_f > 0$, i.e. stochastic fine gradients, we obtain the bound

$$\lim_{k \to \infty} \mathbb{E} \left[ \mathcal{L}(\theta^{(k)}) - \mathcal{L}^\star \right] \leq \frac{\tau^2 \delta \sigma_f^2}{\eta \lambda(\tau)}.$$

We would like to point out the similarity of this estimate to the convergence result for AC/DC (Peste et al., 2021) which corresponds to the SGD case with $\lambda(\tau) = \tau^2$.

While a proof of convergence to the optimal value, or even of the iterates, without the additional assumption of using exact gradients on the fine level would be preferable, this appears to be out of reach unless one is willing to assume (strong) convexity (Bungert et al., 2022), increased smoothness (Zhang & He, 2018), or perform variance reduction (Li et al., 2022). The only results for standard mirror descent in this direction that we are aware of are due to Fatkhullin & He (2024), who show that, in expectation, the best iterate is within one mirror-descent step of an $\varepsilon$-optimal point once sufficiently many steps have been taken and the step-sizes are chosen correctly depending on $\varepsilon$. Their analysis relies on a differentiable mirror map, weak relative convexity of the objective, and a different variant of the PolyakŁojasiewicz inequality.

In Appendix B, we examine convergence in a slightly different setting. Specifically, when the loss function has a finite-sum structure, we show that using a mini-batch approximation as the coarse loss, together with an added linear correction term, makes the multilevel algorithm closely related to standard variance-reduction methods. This connection enables us to establish convergence of the resulting algorithm.

## 5 Numerical Experiments

All gradient estimators in Algorithm 1 are computed using mini-batch approximations. It is also important to note that the ordering of coarse and fine updates is flexible. For instance, one may perform a single fine update – using a mini-batch to update the fine model – followed by $m$ coarse updates. Alternatively, switching can occur at the epoch level, where fine updates are carried out for an entire epoch, and then coarse updates are applied for $m$ consecutive epochs.

As mentioned above, we choose the linear map that selects all non-zero parameters in the current state as the restriction operator. The sparsity of a model is defined to be the percentage of the models' parameters that are zero. We consider promoting unstructured sparsity with standard $\ell_1$-regularization, that is, $J = \lambda \|\cdot\|_1$, and initialize all networks with a sparsity of 99%. Due to our considerations regarding the initialization (see Appendix D), we would need to multiply every sparsified parameter group by $\frac{1}{\sqrt{0.01}} = 10$. However, we instead use a multiplication of 5 as this leads to better results. For training, we use the step-wise switching strategy between fine and coarse model rather than switching epoch-wise, and take $m = 99$. For all experiments, we initialize the learning rate at 0.1 and apply a cosine annealing scheduler.

For ablation studies investigating the effects of the hyperparameters $\lambda$ and $m$ for medium and larger scale experiments we refer to Appendix D

**CIFAR-10 training**  In order to evaluate our proposed training algorithm, we train different neural network architectures on the CIFAR-10 dataset, containing 60,000 32-by-32 color images in 10 different classes. The objective is to construct sparse models that exhibit only marginal reductions in test accuracy.

The hyperparameter $\lambda$ plays a significant role in determining the characteristics of the trained network. As expected, stronger regularization – i.e., larger values of $\lambda$ – leads to increased sparsity in the resulting model,

potentially at the cost of reduced accuracy (see Figure 5a in the Appendix). Therefore, our aim is to find a trade-off between sparsity and accuracy.

We train models using Algorithm 1 with different choices of the regularization parameter $\lambda$ across multiple random seeds to produce models with different levels of sparsity. As baselines, we include models trained with stochastic gradient descent (SGD), LinBreg, RigL (Evci et al., 2020) and pruned versions of the SGD-trained models. The latter were pruned to match the sparsity levels achieved by LinBreg and then fine-tuned. For a fair comparison, the pruned models were trained for only 180 epochs before undergoing an additional 20 epochs of fine-tuning, whereas the baseline methods were trained for the full 200 epochs. Thus, the overall training budget is aligned at 200 epochs across all methods. For models trained with RigL, we used the proposed ErdsRényi-Kernel (ERK) initialization method, as we found it yielded the best results. While our proposed method is robust to different initialization schemes, RigL relies on ERK initialization to achieve competitive results (see Table 2 in Appendix D). We visualize our results in Figure 2 plotting the mean values and standard deviations over multiple random seeds of the test accuracy at the corresponding sparsity levels for different regularizers, for LinBreg, our proposed algorithm, RigL and the pruned and fine-tuned models. Here, we focus on very high levels of sparsity and present the corresponding results. A more comprehensive view across a wider range of sparsity levelsincluding the cases shown hereis provided in Figure 7. We observe that, across all the architectures considered, our method achieves results comparable to those of RigL. Moreover, we emphasize that, while achieving results comparable to LinBreg and pruning on ResNet18, our method requires substantially less gradient information. For the VGG16 architecture, the pruned architectures outperform the other approaches, however, at the cost of requiring training of a full architecture. Moreover, for WideResNet28-10, our algorithm outperforms both LinBreg and the pruning approach by producing models that are sparser and equally accurate. Compared to pruning, our method, RigL (and to some extent also LinBreg) have the advantage of not requiring a dense model to be trained.

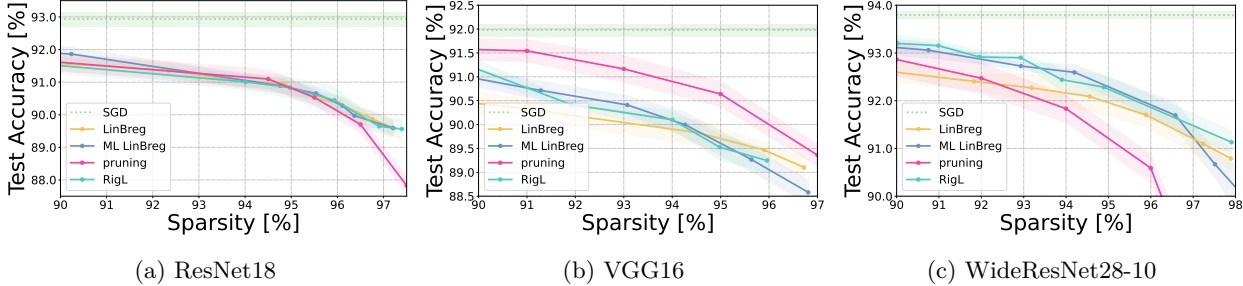

| (a) ResNet18 | (b) VGG16 | (c) WideResNet28-10 |

Figure 2: Achieved accuracy and sparsity of different optimizers on CIFAR-10 for different architectures.

Exploiting sparsity in practice – especially unstructured sparsity as focused on in these experiments – is notoriously difficult, especially on GPUs. To this end, it is common to report theoretical computational savings – see Appendix C for how our method compares to standard training with SGD and LinBreg in this regard. Furthermore, implementations of sparse training algorithms often calculate dense gradients and only simulate sparse training via masking hooks on forward and backward passes. To illustrate that ML LinBreg can lead to real-time savings, we use SparseProp (Nikdan et al., 2023), a package which has implementations of linear and convolutional layers that exploit unstructured sparsity - with the caveat that it is only applicable on CPUs. We train ResNet18 on an AMD Ryzen 5 4500U CPU for 10 epochs where, after each fine update, we replace Torch linear and convolutional layers with SparseProp analogues if doing so leads to faster forward and backward passes. We report the computational performance of SGD and the Torch and SparseProp implementations of ML LinBreg in Figure 3. Utilizing SparseProp reduces training time by 49% compared to the Torch implementation, with time spent for forward and backward passes reduced by 32% and 58%, respectively. In contrast, the Torch variant of ML LinBreg takes slightly longer than SGD since full gradients are still calculated under the hood and the optimizer step has added complexity.

Further CIFAR-10 experiments are reported in Appendix D, including an ablation study investigating the effect of the duration of the coarse period and the regularization strength.

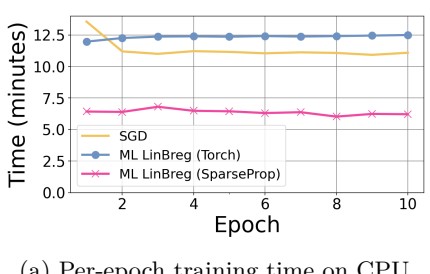

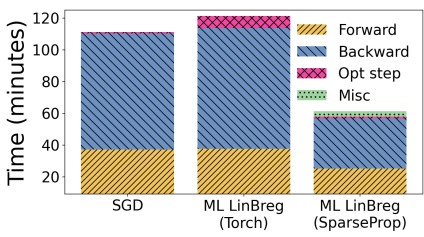

(a) Per-epoch training time on CPU

(b) Breakdown of total training time on CPU

Figure 3: Timing breakdown for training ResNet18 for 10 epochs on an AMD Ryzen 5 4500U CPU when employing SparseProp modules to exploit unstructured sparsity. The sparsity induced by ML LinBreg can lead to real-time computational savings.

**TinyImageNet training** We further evaluate our approach on the TinyImageNet dataset, containing 100,000 64-by-64 color images in 200 different classes, using two different architectures. As in the CIFAR-10 experiments, we train models with the regularizer $J = \lambda \| \cdot \|_1$ for different values of $\lambda$ across multiple random seeds. The results are summarized in Figure 4, where we present the mean values and standard deviations of the achieved test accuracies located at the sparsity of the models. For comparison, we also include a dense baseline trained with standard SGD. We additionally provide the results that can be achieved by pruning and fine-tuning the dense SGD models to different levels of sparsity and by training with RigL and LinBreg. Our findings show that the proposed multilevel method consistently outperforms the standard LinBreg algorithm in our experiments on the TinyImageNet dataset by producing models that are sparser while maintaining, or even improving, test accuracy. Compared to the pruning approach and to RigL, we achieve comparable or better test accuracy across all sparsity levels.

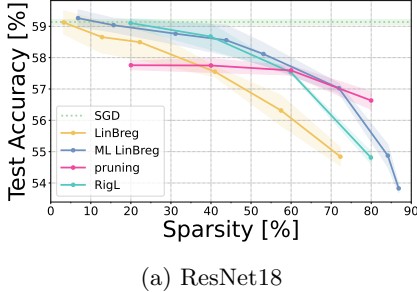

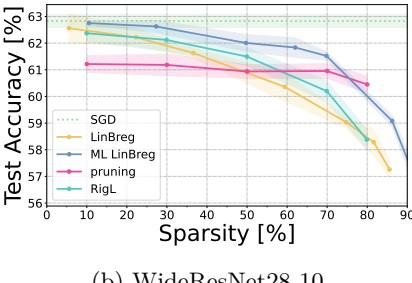

(a) ResNet18

(b) WideResNet28-10

Figure 4: Achieved accuracy and sparsity of different optimizers on TinyImageNet for different architectures.

## 6 Conclusion and Outlook

We proposed a multilevel framework for linearized Bregman iterations in the context of sparse training. We investigated the convergence properties of our method assuming a PL-type condition for the objective and we demonstrated its effectiveness in producing sparse yet accurate models for image classification tasks. An interesting direction for future work is to analyze the convergence of our algorithm in a fully stochastic setting, using gradient estimators in place of exact gradients for both the lower and the higher level problem. Also, relaxing the PL-condition to a local condition of Kurdyka–Łojasiewicz type is an interesting but challenging follow-up problem. Moreover, sparsity-informed training implementations of our method have the potential to reduce training time and resource requirements.

**Reproducibility Statement**

The code used to produce the results is available at anonymous.4open.science/r/SparseLinBreg/.

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

# A  Deferred proofs

In this section, we carry out the proofs for our statements concerning the transfer of subgradients between the coarse and fine level (Propositions 3.1 and 3.2) as well as the descent on the coarse level (Lemma 4.2). In both cases it turns out to be helpful to consider the connection between $\hat{J}_\delta^{(k)} := \frac{1}{2\delta} \| \cdot \|^2 + \hat{J}^{(k)}$, with $\hat{J}^{(k)}$ defined in (10) and $\hat{J} := J_\delta(\theta^{(k)} + P^{(k)}(\cdot - \hat{\theta}^{0,k}))$, both of which are natural ways to restrict the regularizer $J_\delta$ to the coarse level. We observe that these two functions coincide up to an additive constant depending on $\theta^{(k)}$ implying that, in particular, these functions have the same subdifferential and induce the same Bregman divergence. To see that the functions are equal up to a constant, note that

$$(\theta^{(k)} + P^{(k)}(\hat{\theta} - \hat{\theta}^{0,k}))_{(g)} = \begin{cases} \hat{\theta}_{(i)}, & \text{if } g = r^{(k)}(i), \\ (\theta^{(k)})_{(g)}, & \text{otherwise,} \end{cases} \tag{15}$$

due to the definitions of $P^{(k)}$ and $\hat{\theta}^{0,k}$. Using this, it is straightforward to compute

$$\hat{J}(\hat{\theta}) = \frac{1}{2\delta} \|\theta^{(k)} + P^{(k)}(\hat{\theta} - \hat{\theta}^{0,k})\|^2 + J(\theta^{(k)} + P^{(k)}(\hat{\theta} - \hat{\theta}^{0,k}))$$

$$= \frac{1}{2\delta} \|\hat{\theta}\|^2 + \frac{1}{2\delta} \sum_{g \in X} \|(\theta^{(k)})_{(g)}\|^2 + \sum_{i=1}^{G^{(k)}} J_{r^{(k)}(i)}(\hat{\theta}_{(i)}) + \sum_{g \in X} J_g((\theta^{(k)})_{(g)})$$

$$= \hat{J}_\delta^{(k)}(\hat{\theta}) + \text{const}(\theta^{(k)}),$$

where we use the abbreviation

$$X := \{1, 2, \ldots, G\} \setminus r^{(k)}(\{1, 2, \ldots, G^{(k)}\})$$

to denote all groups that are not selected by $R^{(k)}$. Hence, $\hat{J}_\delta^{(k)}$ and $\hat{J}$ indeed coincide up to an additive constant depending on $\theta^{(k)}$.

## A.1  Proofs for Section 3

Having established that $\hat{J}_\delta^{(k)}$ and $\hat{J}$ have the same subdifferential, we can prove Proposition 3.1.

*Proof of Proposition 3.1.* Due to the sum and chain rule for subdifferentials,

$$\partial \hat{J}(\hat{\theta}) = \frac{1}{\delta} \hat{\theta} + (P^{(k)})^T \partial J(\theta^{(k)} + P^{(k)}(\hat{\theta} - \hat{\theta}^{0,k})).$$

for $\hat{\theta} \in \mathbb{R}^{D_k}$ such that $J(\theta^{(k)} + P^{(k)}(\hat{\theta} - \hat{\theta}^{0,k})) < \infty$ and $\partial J(\theta^{(k)} + P^{(k)}(\hat{\theta} - \hat{\theta}^{0,k})) \neq \emptyset$. Consequently, since we have already established that $\hat{J}_\delta^{(k)}$ and $\hat{J}$ share the same subdifferential, for $\hat{\theta} = \hat{\theta}^{0,k}$, we have

$$\partial \hat{J}_\delta^{(k)}(\hat{\theta}^{0,k}) = \partial \hat{J}(\hat{\theta}^{0,k})$$

$$= \frac{1}{\delta} \hat{\theta}^{0,k} + R^{(k)} \partial J(\theta^{(k)})$$

$$= R^{(k)} \partial J_\delta(\theta^{(k)}).$$

Obviously, $\hat{v}^{0,k} = R^{(k)} v^{(k)}$ belongs to the latter, concluding the proof. □

*Proof of Proposition 3.2.* Using (15) with $\hat{\theta} = \hat{\theta}^{m,k}$ together with (7), it is easy to see that

$$\partial_{(g)} J_\delta(\tilde{\theta}^{(k+1)}) = \frac{1}{\delta} \tilde{\theta}^{(k+1)}_{(g)} + \partial J_g((\tilde{\theta}^{(k+1)})_{(g)})$$

$$= \begin{cases} \partial_{(i)} \hat{J}_\delta^{(k)}((\hat{\theta}^{m,k})_{(i)}), & \text{if } g = r^{(k)}(i), \\ \partial_{(g)} J_\delta(\theta^{(k)}), & \text{otherwise.} \end{cases}$$

Since by definition

$$\tilde{v}^{(k+1)} = \begin{cases} (\hat{v}^{m,k})_{(i)}, & \text{if } g = r^{(k)}(i), \\ (v^{(k)})_{(g)}, & \text{otherwise,} \end{cases}$$

together with $\hat{v}^{m,k} \in \partial \hat{J}_\delta^{(k)}(\hat{\theta}^{m,k})$ and $v^{(k)} \in \partial J_\delta(\theta^{(k)})$, we obtain overall that $\tilde{v}^{(k+1)} \in \partial J_\delta(\tilde{\theta}^{(k+1)})$. $\qquad \square$

### A.2  Proof of Lemma 4.2

In this section, we prove Lemma 4.2. Before being able to do so, we show that the coarse objective $\hat{\mathcal{L}}^{(k)}$ is $L$-smooth with respect to the coarse regularizer $\hat{J}_\delta^{(k)}$ if $\mathcal{L}$ is $L$-smooth relative to $J_\delta$. To do so, we will make use of the fact that $\hat{J}_\delta^{(k)}$ and $\hat{J}$ induce the same Bregman divergence.

**Lemma A.1.** *Let Assumption 1 be satisfied. Then, the coarse objective $\hat{\mathcal{L}}^{(k)}$ defined in (9) is $L$-smooth relative to $\hat{J}_\delta^{(k)}$.*

*Proof.* Let $\hat{\theta} \in \mathrm{dom}\, \partial J$ with $\hat{v} \in \partial \hat{J}(\hat{\theta})$ and $\hat{\theta}' \in \mathbb{R}^{D_k}$. We define

$$\tilde{v} := v^{(k)} + P^{(k)}(\hat{v} - \hat{v}^{0,k}),$$
$$\tilde{\theta} := \theta^{(k)} + P^{(k)}(\hat{\theta} - \hat{\theta}^{0,k}),$$

and note that this leads to $\tilde{v} \in \partial J_\delta(\tilde{\theta})$ following from the same argumentation as in the proof of Proposition 3.2. In particular, $\tilde{\theta} \in \mathrm{dom}\, \partial J$. Hence, from the $L$-smoothness of $\mathcal{L}$ relative to $J_\delta$, we deduce

$$\hat{\mathcal{L}}^{(k)}(\hat{\theta}') - \hat{\mathcal{L}}^{(k)}(\hat{\theta}) - \langle \nabla \mathcal{L}(\tilde{\theta}), P^{(k)}(\hat{\theta}' - \hat{\theta}) \rangle \leq L(\hat{J}(\hat{\theta}') - \hat{J}(\hat{\theta}) - \langle \tilde{v}, P^{(k)}(\hat{\theta}' - \hat{\theta}) \rangle)$$

and using $(P^{(k)})^T \nabla \mathcal{L}(\tilde{\theta}) = \nabla \hat{\mathcal{L}}^{(k)}(\hat{\theta})$, we obtain

$$\hat{\mathcal{L}}^{(k)}(\hat{\theta}') - \hat{\mathcal{L}}^{(k)}(\hat{\theta}) - \langle \nabla \hat{\mathcal{L}}^{(k)}(\hat{\theta}), \hat{\theta}' - \hat{\theta} \rangle \leq L(\hat{J}(\hat{\theta}') - \hat{J}(\hat{\theta}) - \langle (P^{(k)})^T \tilde{v}, \hat{\theta}' - \hat{\theta} \rangle).$$

Due to the definition of $\tilde{v}$, we have $(P^{(k)})^T \tilde{v} = R^{(k)} \tilde{v} = \hat{v}$, since $R^{(k)} P^{(k)} = \mathrm{id}$. Consequently,

$$\hat{\mathcal{L}}^{(k)}(\hat{\theta}') - \hat{\mathcal{L}}^{(k)}(\hat{\theta}) - \langle \nabla \hat{\mathcal{L}}^{(k)}(\hat{\theta}), \hat{\theta}' - \hat{\theta} \rangle \leq L D_{\hat{J}}^{\hat{v}}(\hat{\theta}', \hat{\theta}),$$

meaning that $\hat{\mathcal{L}}^{(k)}$ is $L$-smooth relative to $\hat{J}$. Since we have already established that $\hat{J}$ and $\hat{J}_\delta^{(k)}$ induce the same Bregman divergence, relative smoothness with respect to these two functions is equivalent. $\qquad \square$

Having established the relative smoothness of the coarse objective, we are in the position to prove Lemma 4.2.

*Proof of Lemma 4.2.* By Lemma A.1, $\hat{\mathcal{L}}^{(k)}$ is $L$-smooth relative to $\hat{J}_\delta^{(k)}$. Hence, we obtain

$$\hat{\mathcal{L}}^{(k)}(\hat{\theta}^{i+1,k}) \leq \hat{\mathcal{L}}^{(k)}(\hat{\theta}^{i,k}) + \langle \nabla \hat{\mathcal{L}}^{(k)}(\hat{\theta}^{i,k}), \hat{\theta}^{i+1,k} - \hat{\theta}^{i,k} \rangle + L D_{\hat{J}_\delta^{(k)}}^{\hat{v}^{i,k}}(\hat{\theta}^{i+1,k}, \hat{\theta}^{i,k})$$

$$= \hat{\mathcal{L}}^{(k)}(\hat{\theta}^{i,k}) + \langle \hat{g}^{(k)}(\hat{\theta}^{i,k}, \omega), \hat{\theta}^{i+1,k} - \hat{\theta}^{i,k} \rangle$$
$$+ \langle \nabla \hat{\mathcal{L}}^{(k)}(\hat{\theta}^{i,k}) - \hat{g}^{(k)}(\hat{\theta}^{i,k}, \omega), \hat{\theta}^{i+1,k} - \hat{\theta}^{i,k} \rangle + L D_{\hat{J}_\delta^{(k)}}^{\hat{v}^{i,k}}(\hat{\theta}^{i+1,k}, \hat{\theta}^{i,k})$$

$$\leq \hat{\mathcal{L}}^{(k)}(\hat{\theta}^{i,k}) - \frac{1}{\hat{\tau}^{i,k}} D_{\hat{J}_\delta^{(k)}}^{\mathrm{sym}}(\hat{\theta}^{i+1,k}, \hat{\theta}^{i,k}) + \frac{\varepsilon \hat{\tau}^{i,k}}{2} \| \nabla \hat{\mathcal{L}}^{(k)}(\hat{\theta}^{i,k}) - \hat{g}^{(k)}(\hat{\theta}^{i,k}, \omega) \|^2$$
$$+ \frac{1}{2\varepsilon \hat{\tau}^{i,k}} \| \hat{\theta}^{i+1,k} - \hat{\theta}^{i,k} \|^2 + L D_{\hat{J}_\delta^{(k)}}^{\hat{v}^{i,k}}(\hat{\theta}^{i+1,k}, \hat{\theta}^{i,k})$$

$$\leq \hat{\mathcal{L}}^{(k)}(\hat{\theta}^{i,k}) - \frac{2\varepsilon - 2L\varepsilon \hat{\tau}^{i,k} - \delta}{2\varepsilon \hat{\tau}^{i,k}} D_{\hat{J}_\delta^{(k)}}^{\mathrm{sym}}(\hat{\theta}^{i+1,k}, \hat{\theta}^{i,k})$$
$$+ \frac{\varepsilon \hat{\tau}^{i,k}}{2} \| \nabla \hat{\mathcal{L}}^{(k)}(\hat{\theta}^{i,k}) - \hat{g}^{(k)}(\hat{\theta}^{i,k}, \omega) \|^2$$

for any $\varepsilon > 0$ using Young's inequality. Consequently, taking the expectation conditioned on the sigma algebra $\mathcal{F}^{i,k} = \sigma(\mathcal{F}_k, \hat{\theta}^{0,k}, \ldots, \hat{\theta}^{i,k})$, and letting $\varepsilon = \delta$, we arrive at

$$\mathbb{E}\left[\hat{\mathcal{L}}^{(k)}(\hat{\theta}^{i+1,k}) \mid \mathcal{F}^{i,k}\right] \leq \hat{\mathcal{L}}^{(k)}(\hat{\theta}^{i,k}) - \frac{1 - 2L\hat{\tau}^{i,k}}{2\hat{\tau}^{i,k}}\mathbb{E}\left[D_{\hat{J}_\delta^{(k)}}^{\mathrm{sym}}(\hat{\theta}^{i+1,k}, \hat{\theta}^{i,k}) \mid \mathcal{F}^{i,k}\right] + \frac{\delta\hat{\tau}^{i,k}}{2}\sigma_c^2.$$

Thus, for $\hat{\tau}^{i,k} \leq \frac{1}{4L}$, taking the conditional expectation given $\mathcal{F}_k$ and using the tower property, iterating this inequality leads to

$$\mathbb{E}\left[\hat{\mathcal{L}}^{(k)}(\hat{\theta}^{i+1,k}) \mid \mathcal{F}_k\right] \leq \hat{\mathcal{L}}^{(k)}(\hat{\theta}^{0,k}) - \sum_{j=0}^{i} \frac{1}{4\hat{\tau}^{j,k}}\mathbb{E}\left[D_{\hat{J}_\delta^{(k)}}^{\mathrm{sym}}(\hat{\theta}^{j+1,k}, \hat{\theta}^{j,k}) \mid \mathcal{F}_k\right] + \sum_{j=0}^{i} \frac{\delta\sigma_c^2}{2}\hat{\tau}^{j,k}.$$

In particular for $i = m - 1$, we obtain

$$\mathbb{E}\left[\hat{\mathcal{L}}^{(k)}(\hat{\theta}^{m,k}) \mid \mathcal{F}_k\right] \leq \hat{\mathcal{L}}^{(k)}(\hat{\theta}^{0,k}) - \sum_{j=0}^{m-1} \frac{1}{4\hat{\tau}^{j,k}}\mathbb{E}\left[D_{\hat{J}_\delta^{(k)}}^{\mathrm{sym}}(\hat{\theta}^{j+1,k}, \hat{\theta}^{j,k}) \mid \mathcal{F}_k\right] + \sum_{j=0}^{m-1} \frac{\delta\sigma_c^2}{2}\hat{\tau}^{j,k}.$$

Since $\mathcal{L}(\tilde{\theta}^{(k+1)}) = \hat{\mathcal{L}}^{(k)}(\hat{\theta}^{m,k})$ and $\mathcal{L}(\theta^{(k)}) = \hat{\mathcal{L}}^{(k)}(\hat{\theta}^{0,k})$, this yields the desired estimate. $\qquad\square$

## B Convergence in the case of finite sums

In this section, we take a slightly different perspective and demonstrate that multilevel optimization methods can be interpreted as variance reduction methods. To this end, we consider a slightly modified algorithm compared to Algorithm 1. Specifically, we use a different objective function on the coarse level. Moreover, we analyze the resulting algorithm solely from a theoretical perspective without conducting numerical experiments.

For our analysis in this section, we assume that the loss $\mathcal{L}$ has a finite sum structure

$$\mathcal{L}(\theta) = \frac{1}{n}\sum_{j=1}^{n} \mathcal{L}_j(\theta)$$

and each $\mathcal{L}_j$ has a Lipschitz-continuous gradient with Lipschitz-constant $L > 0$. In this setting, we use a mini-batch approximation of the loss as the loss of the coarse level. To be precise, we sample a mini-batch $J^{i,k}$ of size $b < n$ from $[n] \coloneqq \{1, \ldots, n\}$ at the $i$-th coarse step within the $k$-th total cycle in Algorithm 1 and define

$$\hat{\mathcal{L}}^{i,k}(\hat{\theta}) \coloneqq \frac{1}{b}\sum_{j \in J^{i,k}} \mathcal{L}_j(\theta^{(k)} + P^{(k)}(\hat{\theta} - \hat{\theta}^{0,k})).$$

as the coarse loss. As usual in multilevel optimization algorithms (Elshiaty & Petra, 2026; Nash, 2000), we add a linear correction term to this coarse loss to ensure first-order coherence. Precisely, the function, we aim at minimizing in the $i$-th coarse step of the $k$-th total cycle is given by

$$\hat{\Psi}^{i,k}(\hat{\theta}) \coloneqq \hat{\mathcal{L}}^{i,k}(\hat{\theta}) + \langle R^{(k)}\nabla\mathcal{L}(\theta^{(k)}) - \nabla\hat{\mathcal{L}}^{i,k}(\hat{\theta}^{0,k}), \hat{\theta} - \hat{\theta}^{0,k}\rangle. \tag{16}$$

Computing the gradient of this function yields

$$\nabla\hat{\Psi}^{i,k}(\hat{\theta}) = \nabla\hat{\mathcal{L}}^{i,k}(\hat{\theta}) + R^{(k)}\nabla\mathcal{L}(\theta^{(k)}) - \nabla\hat{\mathcal{L}}^{i,k}(\hat{\theta}^{0,k}).$$

In particular, if $\theta^{(k)}$ is a critical point of $\mathcal{L}$, then $\hat{\theta}^{0,k} = R^{(k)}\theta^{(k)}$ is a critical point of $\hat{\Psi}^{i,k}$ for any $i = 1, \ldots, m$. A different perspective on the gradient of the coarse objective is to see it as a variance reduction method. Making use of the special finite sum structure of both $\mathcal{L}$ and $\hat{\mathcal{L}}^{i,k}$, we compute

$$\nabla\hat{\Psi}^{i,k}(\hat{\theta}) = \frac{1}{b}\sum_{j \in J^{i,k}} R^{(k)}\nabla\mathcal{L}_j(\hat{\theta}) + R^{(k)}\frac{1}{n}\sum_{j=0}^{n} \nabla\mathcal{L}_j(\theta^{(k)}) - \frac{1}{b}\sum_{j \in J^{i,k}} R^{(k)}\nabla\mathcal{L}_j(\theta^{(k)}),$$

where we use the abbreviation $\tilde{\theta} := \theta^{(k)} + P^{(k)}(\hat{\theta} - \hat{\theta}^{0,k})$. Thus, $\nabla\hat{\Psi}^{i,k}$ is a combination of current and previous gradients, providing scope for variance reduction. The connection between multilevel optimization and variance reduction has already been investigated, e.g. in Marini et al. (2024); Braglia et al. (2020). We would in particular like to point out the similarity to stochastic variance reduced gradient (SVRG) (Johnson & Zhang, 2013), which uses the same ideas for variance reduction. As before, we denote

$$\hat{\mathcal{L}}^{(k)}(\hat{\theta}) := \mathcal{L}(\theta^{(k)} + P^{(k)}(\hat{\theta} - \hat{\theta}^{0,k})) = \frac{1}{n}\sum_{j=1}^{n}\mathcal{L}_j(\theta^{(k)} + P^{(k)}(\hat{\theta} - \hat{\theta}^{0,k}))$$

as the actual (non-stochastic) objective, we would like to minimize on the coarse level. We observe that $\nabla\hat{\Psi}^{i,k}(\hat{\theta})$ is an unbiased estimator of $\nabla\hat{\mathcal{L}}^{(k)}(\hat{\theta})$ and we can prove a variance bound.

**Lemma B.1.** $\nabla\hat{\Psi}^{i,k}(\hat{\theta})$ *is an unbiased estimator of* $\nabla\hat{\mathcal{L}}^{(k)}(\hat{\theta})$, *i.e.*

$$\mathbb{E}\left[\nabla\hat{\Psi}^{i,k}(\hat{\theta})\right] = \nabla\hat{\mathcal{L}}^{(k)}(\hat{\theta})$$

*and satisfies the variance bound*

$$\mathbb{E}\left[\|\nabla\hat{\Psi}^{i,k}(\hat{\theta}) - \nabla\hat{\mathcal{L}}^{(k)}(\hat{\theta})\|^2\right] \le \frac{L^2(n-b)}{b(n-1)}\|\hat{\theta} - \hat{\theta}^{0,k}\|^2,$$

*where all expected values are taken with respect to the mini-batch sampling.*

*Proof.* Taking the expected value with respect to drawing the mini-batch, we can easily compute

$$\mathbb{E}\left[\nabla\hat{\Psi}^{i,k}(\hat{\theta})\right] = R^{(k)}\nabla\mathcal{L}(\theta^{(k)} + P^{(k)}(\hat{\theta} - \hat{\theta}^{0,k})) = \nabla\hat{\mathcal{L}}^{(k)}(\hat{\theta}).$$

Moreover, turning towards the variance, we observe that

$$\mathbb{E}\left[\|\nabla\hat{\Psi}^{i,k}(\hat{\theta}) - \nabla\hat{\mathcal{L}}^{(k)}(\hat{\theta})\|^2\right] = \mathbb{E}\left[\|\nabla\hat{\mathcal{L}}^{i,k}(\hat{\theta}) - \nabla\hat{\mathcal{L}}^{i,k}(\hat{\theta}^{0,k}) - (\nabla\hat{\mathcal{L}}^{(k)}(\hat{\theta}) - \nabla\hat{\mathcal{L}}^{(k)}(\hat{\theta}^{0,k}))\|^2\right]$$

$$= \mathbb{E}\left[\|\frac{1}{b}\sum_{j\in J^{i,k}}R^{(k)}(\nabla\mathcal{L}_j(\tilde{\theta}) - \nabla\mathcal{L}_j(\theta^{(k)})) - (\nabla\hat{\mathcal{L}}^{(k)}(\hat{\theta}) - \nabla\hat{\mathcal{L}}^{(k)}(\hat{\theta}^{0,k}))\|^2\right]$$

using the abbreviation $\tilde{\theta} := \theta^{(k)} + P^{(k)}(\hat{\theta} - \hat{\theta}^{0,k})$. In the case $b = 1$, we can estimate the variance by the second moment and arrive at

$$\mathbb{E}_{j\sim\text{Unif}(1,...,n)}\left[\|R^{(k)}(\nabla\mathcal{L}_j(\tilde{\theta}) - \nabla\mathcal{L}_j(\theta^{(k)})) - (\nabla\hat{\mathcal{L}}^{(k)}(\hat{\theta}) - \nabla\hat{\mathcal{L}}^{(k)}(\hat{\theta}^{0,k}))\|^2\right]$$

$$\le \mathbb{E}_{j\sim\text{Unif}(1,...,n)}\left[\|R^{(k)}(\nabla\mathcal{L}_j(\theta^{(k)} + P^{(k)}(\hat{\theta} - \hat{\theta}^{0,k})) - \nabla\mathcal{L}_j(\theta^{(k)}))\|^2\right]$$

$$\le \|R^{(k)}\|^2 L^2\|\theta^{(k)} + P^{(k)}(\hat{\theta} - \hat{\theta}^{0,k}) - \theta^{(k)}\|^2$$

$$\le L^2\|\hat{\theta} - \hat{\theta}^{0,k}\|^2,$$

using the Lipschitz continuity of $\nabla\mathcal{L}_j$ and the fact that $\|R^{(k)}\|, \|P^{(k)}\| \le 1$. In the case $b > 1$, since we draw without replacement, the variance scales with $\frac{n-b}{b(n-1)}$, leading to

$$\mathbb{E}\left[\|\nabla\hat{\Psi}^{i,k}(\hat{\theta}) - \nabla\hat{\mathcal{L}}^{(k)}(\hat{\theta})\|^2\right] \le \frac{L^2(n-b)}{b(n-1)}\|\hat{\theta} - \hat{\theta}^{0,k}\|^2.$$

$\square$

In this setting, the linearized Bregman iteration scheme on the coarse level reads

$$\hat{g}^{i,k} = \nabla\hat{\Psi}^{i,k},$$
$$\hat{v}^{i+1,k} = \hat{v}^{i,k} - \hat{\tau}\hat{g}^{i,k},$$
$$\hat{\theta}^{i+1,k} = \text{prox}_{\delta\hat{J}^{(k)}}(\delta\hat{v}^{i+1,k}).$$

We investigate the descent of $\hat{\mathcal{L}}^{(k)}$ during the coarse level iterations and assume that the number of steps on the coarse level $m$ is at least 2. Note that for our setup to make sense, we require the full gradient of the loss at $\theta^{(k)}$ and therefore $m = 1$ would correspond to no stochasticity at all.

**Lemma B.2.** *If the step size $\hat{\tau}$ is chosen such that*

$$\hat{\tau} \leq \min\left\{\frac{1}{2L\delta}, \sqrt{\frac{b(n-1)}{2\delta^2 L^2(n-b)m(m-1)}}\right\},$$

*then*

$$\mathbb{E}\left[\mathcal{L}(\tilde{\theta}^{(k+1)}) \,|\, \mathcal{F}_k\right] \leq \mathcal{L}(\theta^{(k)}) - \frac{1}{\hat{\tau}}\sum_{j=1}^{m}\mathbb{E}\left[D_{\hat{\jmath}^{(k)}}^{\mathrm{sym}}(\hat{\theta}^{j,k}, \hat{\theta}^{j-1,k}) \,|\, \mathcal{F}_k\right] - \frac{1}{8\delta\hat{\tau}}\sum_{j=1}^{m}\mathbb{E}\left[\|\hat{\theta}^{j,k} - \hat{\theta}^{j-1,k}\|^2 \,|\, \mathcal{F}_k\right].$$

*Proof.* Using the $L$-smoothness of $\hat{\mathcal{L}}^{(k)}$ with respect to $\frac{1}{2}\|\cdot\|^2$, which follows from the $L$-smoothness with respect to $\frac{1}{2}\|\cdot\|^2$ of each $\mathcal{L}_j$ (see the descent lemma for $L$-smooth functions (Bauschke & Combettes, 2011; Beck, 2017) and Lemma A.1), yields

$$\hat{\mathcal{L}}^{(k)}(\hat{\theta}^{i+1,k}) - \hat{\mathcal{L}}^{(k)}(\hat{\theta}^{i,k}) \leq \langle \nabla\hat{\mathcal{L}}^{(k)}(\hat{\theta}^{i,k}), \hat{\theta}^{i+1,k} - \hat{\theta}^{i,k}\rangle + \frac{L}{2}\|\hat{\theta}^{i+1,k} - \hat{\theta}^{i,k}\|^2$$

$$= \langle \nabla\hat{\mathcal{L}}^{(k)}(\hat{\theta}^{i,k}) - \hat{g}^{i,k}, \hat{\theta}^{i+1,k} - \hat{\theta}^{i,k}\rangle + \langle \hat{g}^{i,k}, \hat{\theta}^{i+1,k} - \hat{\theta}^{i,k}\rangle + \frac{L}{2}\|\hat{\theta}^{i+1,k} - \hat{\theta}^{i,k}\|^2$$

$$\leq \frac{\hat{\tau}}{2\varepsilon}\|\nabla\hat{\mathcal{L}}^{(k)}(\hat{\theta}^{i,k}) - \hat{g}^{i,k}\|^2 - \frac{1}{\hat{\tau}}D_{\hat{\jmath}_\delta^{(k)}}^{\mathrm{sym}}(\hat{\theta}^{i+1,k}, \hat{\theta}^{i,k}) + \frac{L\hat{\tau} + \varepsilon}{2\hat{\tau}}\|\hat{\theta}^{i+1,k} - \hat{\theta}^{i,k}\|^2.$$

for all $\varepsilon > 0$. Taking the expectation conditioned on $\mathcal{F}^{i,k} = \sigma(\mathcal{F}_k, \hat{\theta}^{0,k}, \ldots, \hat{\theta}^{i,k})$ and using the variance bound established in Lemma B.1, we obtain

$$\mathbb{E}\left[\hat{\mathcal{L}}^{(k)}(\hat{\theta}^{i+1,k}) \,|\, \mathcal{F}^{i,k}\right] \leq \hat{\mathcal{L}}^{(k)}(\hat{\theta}^{i,k}) + \frac{\hat{\tau}L^2(n-b)}{2\varepsilon b(n-1)}\|\hat{\theta}^{i,k} - \hat{\theta}^{0,k}\|^2 - \frac{1}{\hat{\tau}}\mathbb{E}\left[D_{\hat{\jmath}_\delta^{(k)}}^{\mathrm{sym}}(\hat{\theta}^{i+1,k}, \hat{\theta}^{i,k}) \,|\, \mathcal{F}^{i,k}\right]$$

$$+ \frac{L\hat{\tau} + \varepsilon}{2\hat{\tau}}\mathbb{E}\left[\|\hat{\theta}^{i+1,k} - \hat{\theta}^{i,k}\|^2 \,|\, \mathcal{F}^{i,k}\right]$$

$$\leq \hat{\mathcal{L}}^{(k)}(\hat{\theta}^{i,k}) + \frac{\hat{\tau}L^2(n-b)}{2\varepsilon b(n-1)}i\sum_{j=1}^{i}\|\hat{\theta}^{j,k} - \hat{\theta}^{j-1,k}\|^2$$

$$- \frac{1}{\hat{\tau}}\mathbb{E}\left[D_{\hat{\jmath}_\delta^{(k)}}^{\mathrm{sym}}(\hat{\theta}^{i+1,k}, \hat{\theta}^{i,k}) \,|\, \mathcal{F}^{i,k}\right] + \frac{L\hat{\tau} + \varepsilon}{2\hat{\tau}}\mathbb{E}\left[\|\hat{\theta}^{i+1,k} - \hat{\theta}^{i,k}\|^2 \,|\, \mathcal{F}^{i,k}\right].$$

We take the expectation conditioned on $\mathcal{F}_k$ and iterate this inequality to arrive at

$$\mathbb{E}\left[\hat{\mathcal{L}}^{(k)}(\hat{\theta}^{i+1,k}) \,|\, \mathcal{F}_k\right] \leq \mathbb{E}\left[\hat{\mathcal{L}}^{(k)}(\hat{\theta}^{0,k}) \,|\, \mathcal{F}_k\right] + \frac{\hat{\tau}L^2(n-b)}{2\varepsilon b(n-1)}\frac{i(i+1)}{2}\sum_{j=1}^{i}\mathbb{E}\left[\|\hat{\theta}^{j,k} - \hat{\theta}^{j-1,k}\|^2 \,|\, \mathcal{F}_k\right]$$

$$- \frac{1}{\hat{\tau}}\sum_{j=1}^{i+1}\mathbb{E}\left[D_{\hat{\jmath}_\delta^{(k)}}^{\mathrm{sym}}(\hat{\theta}^{j,k}, \hat{\theta}^{j-1,k}) \,|\, \mathcal{F}_k\right] + \frac{L\hat{\tau} + \varepsilon}{2\hat{\tau}}\sum_{j=1}^{i+1}\mathbb{E}\left[\|\hat{\theta}^{j,k} - \hat{\theta}^{j-1,k}\|^2 \,|\, \mathcal{F}_k\right].$$

In particular, for $i + 1 = m$, we obtain

$$\mathbb{E}\left[\hat{\mathcal{L}}^{(k)}(\hat{\theta}^{m,k}) \,|\, \mathcal{F}_k\right] \leq \mathbb{E}\left[\hat{\mathcal{L}}^{(k)} \,|\, \mathcal{F}_k\right] - \frac{1}{\hat{\tau}}\sum_{j=1}^{m}\mathbb{E}\left[D_{\hat{\jmath}^{(k)}}^{\mathrm{sym}}(\hat{\theta}^{j,k}, \hat{\theta}^{j-1,k}) \,|\, \mathcal{F}_k\right]$$

$$- \frac{2b\varepsilon(n-1)(2 - \delta\varepsilon - L\delta\hat{\tau}) - \hat{\tau}^2 L^2(n-b)m(m-1)\delta}{4\varepsilon b(n-1)\delta\hat{\tau}}\sum_{j=1}^{m}\mathbb{E}\left[\|\hat{\theta}^{j,k} - \hat{\theta}^{j-1,k}\|^2 \,|\, \mathcal{F}_k\right]$$

Considering this fraction, we observe that for $\varepsilon = \frac{1}{\delta}$ and $\hat{\tau} \le \frac{1}{2L\delta}$, we have

$$\frac{2b\varepsilon(n-1)(2-\delta\varepsilon-L\delta\hat{\tau})-\hat{\tau}^2L^2(n-b)m(m-1)\delta}{4\varepsilon b(n-1)\delta\hat{\tau}} = \frac{\frac{2b}{\delta}(n-1)(1-L\delta\hat{\tau})-\hat{\tau}^2L^2(n-b)m(m-1)\delta}{4b(n-1)\hat{\tau}}$$

$$\ge \frac{\frac{b}{\delta}(n-1)-\hat{\tau}^2L^2(n-b)m(m-1)\delta}{4b(n-1)\hat{\tau}}.$$

Moreover, due to the bound for the stepsize, we have

$$\frac{\frac{b}{\delta}(n-1)-\hat{\tau}^2L^2(n-b)m(m-1)\delta}{4b(n-1)\hat{\tau}} \ge \frac{1}{8\delta\hat{\tau}}.$$

Thus, with $\hat{\mathcal{L}}^{(k)}(\hat{\theta}^{m,k}) = \mathcal{L}(\tilde{\theta}^{(k+1)})$ and $\hat{\mathcal{L}}^{(k)}(\hat{\theta}^{0,k}) = \mathcal{L}(\theta^{(k)})$ we obtain the desired result. $\qquad\square$

Combining Lemma B.2 and Lemma 4.1 yields the following convergence statement with constant step-sizes on the coarse level.

**Theorem B.3.** *Let Assumption 2 be satisfied and the step sizes chosen such that*

$$\tau \le \frac{1}{L} \quad \text{and} \quad \hat{\tau} \le \min\left\{\frac{1}{2L\delta}, \sqrt{\frac{b(n-1)}{2\delta^2L^2(n-b)m(m-1)}}\right\}.$$

*Let* $(\theta^{(k)})$ *be the sequence generated by Algorithm 1 with exact gradients on the fine level and the coarse objective defined in* (16). *Then,*

$$\mathbb{E}\left[\mathcal{L}(\theta^{(k)})-\mathcal{L}^\star\right] \le (1-r)^k(\mathcal{L}(\theta^{(0)})-\mathcal{L}^\star)-\sum_{i=0}^{k-1}(1-r)^{k-i}\hat{\rho}_i,$$

*where* $r = \frac{\eta\lambda(\tau)}{\tau}$ *and*

$$\hat{\rho}_i = \frac{1}{\hat{\tau}}\sum_{j=1}^m \mathbb{E}\left[D_{\hat{j}^{(k)}}^{\text{sym}}(\hat{\theta}^{j,k},\hat{\theta}^{j-1,k})\right] + \frac{1}{8\delta\hat{\tau}}\sum_{j=1}^m \mathbb{E}\left[\|\hat{\theta}^{j,k}-\hat{\theta}^{j-1,k}\|^2\right].$$

## C Computational savings

Regarding the computational effort, we follow Evci et al. (2020) and estimate the theoretical FLOPs required for training. As in their approach, we approximate the FLOPs for one training step by summing the forward pass FLOPs and twice the forward pass FLOPs for the backward pass. We only account for the FLOPs of convolutional and linear layers. Other operations such as BatchNorm, pooling, and residual additions are ignored, as their contribution to the total FLOPs is negligible compared to the dominant convolutional and linear computations. Denoting the FLOPs of a sparse forward pass by $f_S$ and of a dense forward pass by $f_D$, the expected FLOPs per training step are

$$\frac{m\cdot 3f_S+(2f_S+f_D)}{m+1},$$

since every $(m+1)$-th step computes a full gradient. This matches the estimation used in RigL, where the network structure is fixed for most iterations and the full gradient is only computed periodically. In our method, however, the sparsity level evolves during training, whereas RigL enforces a fixed sparsity. As a result, particularly in the early stages of training, our method requires substantially fewer FLOPs. By contrast, LinBreg incurs $2f_S+f_D$ FLOPs per training step, since a full gradient is computed at every iteration.

We estimate the theoretical training FLOPs for the WideResNet28-10 models reported in Table 1. Using standard SGD training as the dense baseline, our analysis indicates that LinBreg requires only 0.38× the FLOPs of the baseline, while our method further reduces this to 0.06×.

For the VGG16 models from Table 1, LinBreg requires $0.47\times$ the FLOPs of the dense model, whereas our method only needs $0.18\times$. Finally, for ResNet18, we consider the LinBreg approach with $\lambda = 0.2$, which leads to 96.03% sparsity on average, and the ML LinBreg with $\lambda = 0.007$, which reaches 96.12% sparsity. This results in a theoretical FLOPs reduction of $0.43\times$ for the former and $0.12\times$ for the latter.

## D   Background on numerical experiments

**Initialization**   As previously mentioned, all networks are initialized in a highly sparse fashion, allowing the training algorithm to subsequently activate additional parameters by setting them to non-zero values. For this initialization, we follow the approach proposed by Bungert et al. (2022).

He et al. (2015) suggest that for ReLU activation functions, the variance of the weights should satisfy

$$\mathrm{Var}\left[W^l\right] = \frac{2}{n_{l-1}}, \tag{17}$$

where $n_l$ denotes the size of the $l$-th layer.

To obtain a sparse initialization, we apply binary masks to dense weight matrices, defining the initial weights as

$$W^l = \tilde{W}^l \odot M^l,$$

where $\odot$ denotes the element-wise (Hadamard) product, and each entry of the mask $M^l$ is independently drawn from a Bernoulli distribution with parameter $r \in [0,1]$. This construction implies that the variance of the resulting weights scales as

$$\mathrm{Var}\left[W^l\right] = r \, \mathrm{Var}\left[\tilde{W}^l\right].$$

To ensure that the condition in (17) is met, we adjust the distribution of $\tilde{W}^l$ accordingly. For instance, if $\tilde{W}^l$ originally satisfies (17), we scale it accordingly.

**CIFAR-10 training**   We summarize some of the results from our training on the CIFAR-10 dataset in Table 1, most of which are also visualized in Figure 2. As previously mentioned, we use $J = \lambda\|\cdot\|_1$ as the regularizer and choose to perform a full update on the fine level every 100 iterations, meaning that $m = 99$ in Algorithm 1.

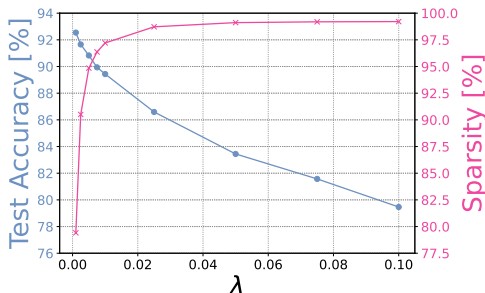
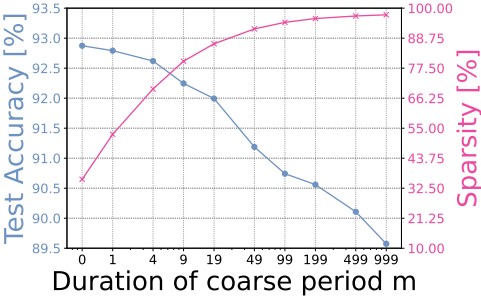

(a) Accuracy and sparsity for ML LinBreg with different values of $\lambda$ with fixed $m$

(b) Accuracy and sparsity for ML LinBreg with different values of $m$ with fixed $\lambda$

Figure 5: Ablation study comparing accuracy and sparsity across different choices of the hyperparameters $\lambda$ and $m$ for ResNet18 on CIFAR-10.

To demonstrate the effect of the regularization parameter $\lambda$ in our training procedure, we train models with varying values of $\lambda$ and evaluate both the test accuracy and sparsity of the resulting models. As expected,

smaller values of $\lambda$ yield highly accurate but less sparse models, whereas increasing $\lambda$ results in reduced accuracy but greater sparsity. The results of this ablation study are presented in Figure 5a.

In Figure 5b, we perform an ablation study to investigate the effect of the coarse update duration $m$ in Algorithm 1 on test accuracy and sparsity of the trained models. All other parameters are fixed; in particular, the regularizer is chosen as $J = 0.005\|\cdot\|_1$. The network is trained on multiple random seeds, and we report the mean values in the figure. Note that the value $m = 0$ corresponds to the case without coarse updates, i.e., the standard LinBreg algorithm. As expected, increasing $m$ – and thus prolonging the freezing period – quickly yields sparser models, where the reduction in accuracy is minor.

Table 1: Total sparsity and test accuracy for different network architectures and different optimizers

| Architecture | Optimizer | Sparsity [%] | Test acc [%] |
|---|---|---|---|
| ResNet18 | SGD | $0.39 \pm 0.08$ | $92.93 \pm 0.23$ |
| | Prune+Fine-Tuning | $95.00$ | $90.84 \pm 0.30$ |
| | ML LinBreg ($\lambda = 0.005$) | $94.76 \pm 0.12$ | $90.89 \pm 0.21$ |
| | ML LinBreg ($\lambda = 0.007$) | $96.12 \pm 0.08$ | $90.24 \pm 0.33$ |
| | ML LinBreg ($\lambda = 0.01$) | $97.20 \pm 0.06$ | $89.60 \pm 0.27$ |
| | LinBreg ($\lambda = 0.2$) | $96.03 \pm 0.04$ | $90.35 \pm 0.22$ |
| VGG16 | SGD | $0.11 \pm 0.02$ | $91.98 \pm 0.13$ |
| | Prune+Fine-Tuning | $92.00$ | $91.39 \pm 0.18$ |
| | LinBreg ($\lambda = 0.1$) | $91.82 \pm 0.05$ | $90.21 \pm 0.24$ |
| | ML LinBreg ($\lambda = 0.003$) | $91.29 \pm 0.18$ | $90.71 \pm 0.19$ |
| WideResNet28-10 | SGD | $0.17 \pm 0.03$ | $93.79 \pm 0.08$ |
| | Prune+Fine-Tuning | $96.00$ | $90.55 \pm 0.32$ |
| | LinBreg ($\lambda = 0.1$) | $95.89 \pm 0.16$ | $91.70 \pm 0.25$ |
| | ML LinBreg ($\lambda = 0.005$) | $96.58 \pm 0.14$ | $91.69 \pm 0.27$ |

To obtain some additional information on the training procedure, we track the mean and standard deviation of the validation accuracy and the sparsity as well as the train loss throughout the training procedure. The resulting plots for ResNet18 are displayed in Figure 6. We observe that freezing the network structure neither affects the final characteristics of the model nor alters the training speed.

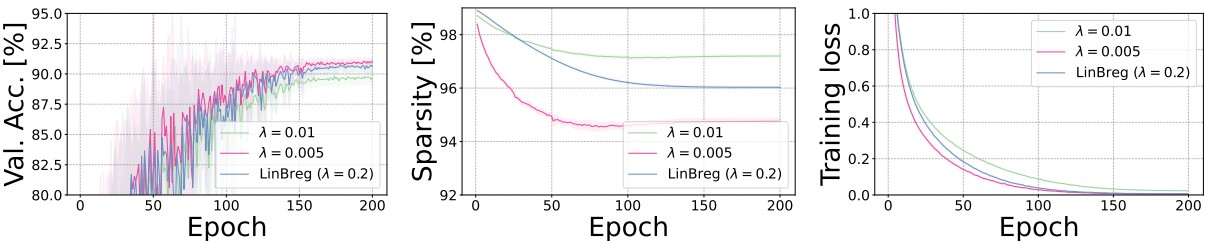

Figure 6: Mean and standard deviation of validation accuracy, sparsity, and train loss of the model parameters over training epochs, shown for different regularization parameters, $\lambda = 0.005$ and $\lambda = 0.01$ and for LinBreg with $\lambda = 0.2$.

Moreover, in addition to comparing the different algorithms for extremely sparse models in Figure 2, we also compare them over the full range of possible levels of sparsity in Figure 7. As before, we include SGD as a dense baseline, as well as pruned versions of the SGD-trained models at various levels of sparsity. For sparsity levels that are not as extreme as those considered in Figure 2, we observe that our method is at least as performant as LinBreg and the pruning approach across different network architectures.

**Mask initialization**   We explore the effect of the mask initialization scheme for CIFAR-10 ResNet18 models trained by both ML LinBreg and RigL (Evci et al., 2020). In particular, we consider three mask initialization schemes as proposed in (Evci et al., 2020).

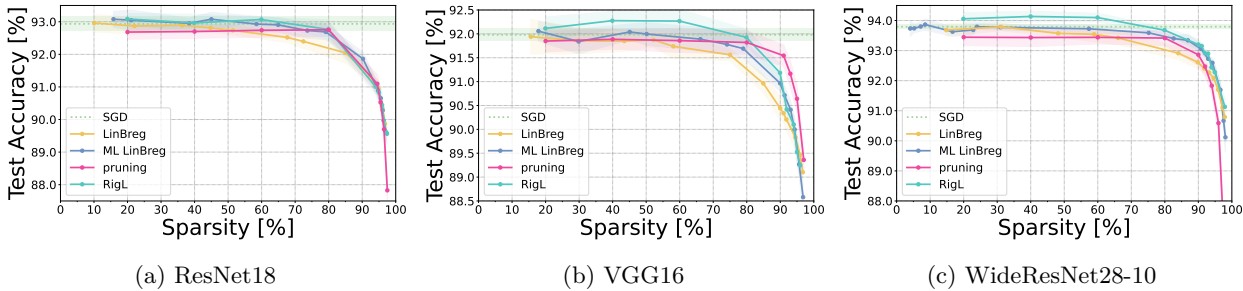

Figure 7: Accuracy and sparsity for different regularizers with LinBreg and ML LinBreg as well as a pruning baseline on CIFAR-10 for different architectures

- *Uniform:* The mask for each layer of the network is uniformly sampled to achieve the target initial sparsity, leading to the same level of sparsity in each layer.

- *ErdsRényi (ER):* Both linear and convolutional layers have sparsities proportional to scores

$$\frac{n_{in} + n_{out}}{n_{in} n_{out}}$$

where $n_{in}/n_{out}$ refers to the dimensions of the input/output of a linear layer and the number of input/output channels for a convolutional layer.

- *ErdsRényi-Kernel (ERK):* Similar to ER except the scores for convolutional layers with $k \times k$ kernels are given by

$$\frac{n_{in} + n_{out} + 2k}{n_{in} n_{out} k^2}.$$

In the case of ER and ERK, one would in practice calculate the scores and then determine a global rescaling factor of said scores such that the determined masked model achieves the specified target sparsity. The motivation of ER is that layers with many parameters should be very sparse, and ERK makes this more explicit in the case of convolutional layers.

We remark that while the variance-preserving rescaling factor $r$ (see Appendix D) will be constant across layers when using a uniform mask initialization, for ER and ERK one must adjust the value of $r$ according to the layer's determined sparsity.

We report the achieved sparsity and test accuracy of RigL (target sparsity of 95%) and ML LinBreg ($\lambda = 0.005$) for different mask initialization schemes in Table 2. We see that, when using the variance-preserving rescaling for the weight initialization, the mask initialization can vary the accuracy of models trained via RigL by 2.5% whereas ML LinBreg is far more robust to the initialization and varies by only 0.6%.

Table 2: Performance of RigL and ML LinBreg for various mask initialization schemes across 5 seeds.

| Optimizer | Mask initialization | Sparsity [%] | Test acc [%] |
|---|---|---|---|
| | Uniform | 94.918 | 88.280 ś 0.14 |
| RigL | ER | 94.918 | 88.274 ś 0.40 |
| | ERK | 94.975 | 90.830 ś 0.32 |
| | Uniform | 94.687 ś 0.10 | 90.812 ś 0.33 |
| ML LinBreg | ER | 94.788 ś 0.06 | 91.448 ś 0.15 |
| | ERK | 94.835 ś 0.05 | 91.180 ś 0.23 |

**Structured sparsity** Even though, we do not induce any structured sparsity through the regularizer $J$, we observe that for our trained models many of the 2D-kernels are zero. To measure this kind of sparsity, we

define the convolutional sparsity as

$$S_{\text{conv}} := 1 - \frac{\sum_{l \in \mathcal{I}_{\text{conv}}} |\{K_{ij}^l \neq 0\}|}{\sum_{l \in \mathcal{I}_{\text{conv}}} c_{l-1} \cdot c_l},$$

where $\mathcal{I}_{\text{conv}}$ denotes the index set of all convolutional layers, $c_{l-1}$ and $c_l$ are the number of input and output channels of layer $l \in \mathcal{I}_{\text{conv}}$, respectively, and $K_{ij}^l$ is the kernel connecting input channel $i$ to output channel $j$.

For example, in the WideResNet28-10 case with $\lambda = 0.005$, where we observed a test accuracy of $91.69\% \pm 0.27\%$ we obtain $S_{\text{conv}} = 78.21\% \pm 0.82\%$, meaning that almost 80% of input-output connections are zero. We can further increase this structured sparsity by taking the group $\ell_{1,2}$ norm from (8) as the regularizer. We again scale it by some positive factor $\lambda$ to control its influence on the optimization procedure. With this approach, we achieve a convolutional sparsity of $S_{\text{conv}} = 84.06\% \pm 0.79\%$ while maintaining a test accuracy of $91.45\% \pm 0.59\%$. By increasing the strength of the regularizer, this can be further improved to $93.09\% \pm 0.46\%$ convolutional sparsity, with a corresponding test accuracy of $90.48\% \pm 0.72\%$.

**Scaling up**   To investigate how increasing the scale of the experiments affects the results when keeping the hyperparameters fixed, we generated the same plots as in Figure 5, this time for the WideResNet28-10 architecture on the TinyImageNet dataset. This setup simultaneously increases both the network capacity and the size and complexity of the dataset. The results of this analysis are shown in Figure 8.

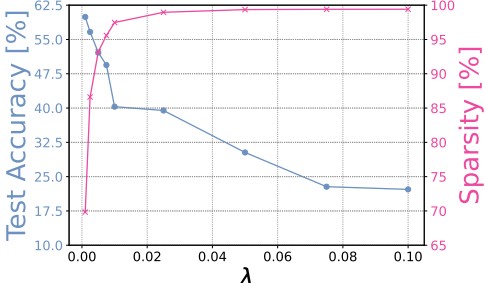
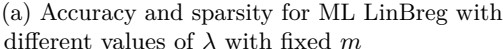
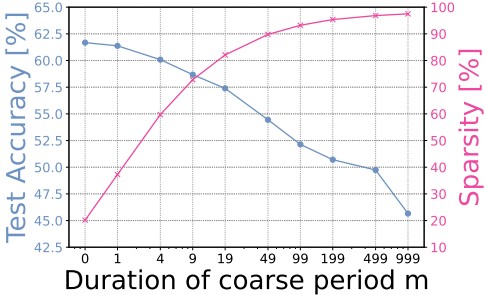

(a) Accuracy and sparsity for ML LinBreg with different values of $\lambda$ with fixed $m$

(b) Accuracy and sparsity for ML LinBreg with different values of $m$ with fixed $\lambda$

Figure 8: Ablation study comparing accuracy and sparsity across different choices of the hyperparameters $\lambda$ and $m$ for WideResNet28-10 on TinyImageNet.

Although the sparsity and test accuracy values are, of course, not identical to those obtained on CIFAR-10, the sparsity levels achieved under the exact same hyperparameters remain in a similar range. As expected, the test accuracies are not directly comparable, since the classification task on TinyImageNet with 200 classes is substantially more challenging than CIFAR-10 with only 10 classes. Furthermore, we observe that small values of $\lambda \approx 0.005$ give best result for both the small scale and the large scale experiment, suggesting that this parameter is pretty scale-invariant. On the other hand, it appears like the large scale experiment yields better results for smaller values of $m$ than the small scale one which is potentially also due to the increased difficulty of the learning problem requiring more exact gradient information.

