# OpenReview forum: "Sparse Training of Neural Networks based on Multilevel Mirror Descent"
_TMLR — Under review for TMLR_

### Review · Reviewer_JLdH · 2026-06-17

**Summary Of Contributions:**

The paper proposes Multilevel LinBreg (ML LinBreg), which extends the LinBreg sparse training algorithm by periodically freezing the non-zero support and restricting updates to active parameters, framed as a two-level multilevel optimization scheme. The coarse level runs $m$ LinBreg steps over active parameters; the fine level performs a full update every $m+1$ steps. A convergence result is proved under relative smoothness and a PL-type inequality for a simplified version of the algorithm (exact gradients in the fine and stochastic gradients in the coarse regime). Experiments on CIFAR-10 and TinyImageNet with ResNet18, VGG16, and WideResNet28-10 are presented.

**Strengths:**
- The paper is generally well-written and well-organized,
- Some theoretical results are provided,
- Ablations over $\lambda$ and $m$,
- Code is provided,
- Multiple seeds, standard deviations reported throughout

**Weaknesses:**
- The theoretical contribution is weaker than presented. The paper claims to prove a "sublinear convergence result," but no convergence rate is provided. Theorem 4.4 gives a recursion with unresolved residual terms, and Corollary 4.5 gives only asymptotic convergence under $\hat\tau^{(k)}\to 0$. Additionally, the theory gives no reason to prefer $m>0$ over $m=0$ (i.e., standard LinBreg from prior work). Assumption 2 is only verified in the Euclidean case.
- The empirical comparisons are methodologically questionable (hyperparameters were not tuned), which could make the comparisons unreliable.
- Several claimed improvements are not supported by the paper's own data. The proposed method is generally interleaved with baselines rather than outperforming them, and is worse than RigL in most experiments.
- The paper states the proofs are "very similar" to Bauschke et al. (2019) and Elshiaty \& Petra (2026) with the main theorem structure adapted from prior work. The authors do not clearly explain where the technical novelty lies.

**Additional Comments:**

- Minor presentation issue: The symmetrized Bregman divergence is used in the proof of Lemma 4.1 before being formally defined (equation (13) appears after its use in the proof).

- The paper writes "$D^p_J(\theta, \theta) = 0$ and, due to convexity of $J$ $D^p_J(\tilde\theta, \theta) \geq 0$" as though non-negativity is an unconditional consequence of convexity. It is not: non-negativity holds only when $p \in \partial J(\theta)$. The algorithms maintain this invariant by construction, so the divergence is non-negative wherever it is actually used in the paper, but the statement as written is imprecise and should be corrected.

**Audience:**

Yes

**Audience Explanation:**

Yes. Sparse training is an active area with a clear practical motivation, and the combination of Bregman iterations with a multilevel framework that periodically freezes the active support is an interesting idea. The results may interest readers working on optimization theory and neural network compression.

**Broader Impact Concerns:**

No explicit broader impact statement needed for this type of work.

**Claims And Evidence:**

No

**Claims Explanation:**

Many of the paper's conclusions are supported by the results. However, some claims seem misleading/incorrect/overstated relative to the available theoretical and empirical evidence:

1. Corollary 4.5 shows that the method achieves convergence. This statement is correct but substantially incomplete. The paper describes its contribution as a "sublinear convergence result," but no convergence rate is given. Theorem 4.4 provides a recursion whose residual terms $\hat\rho_i$ contain both a positive noise contribution and a negative Bregman descent contribution that are not resolved into a closed-form bound. Corollary 4.5 establishes only asymptotic convergence under the condition $\hat\tau^{(k)}\to 0$. Without an explicit rate, the theory cannot be compared to any other algorithm. It also gives no theoretical reason to prefer $m>0$ over $m=0$ (when $m=0$ the algorithm reduces to standard LinBreg from prior work, and the theorem is silent on whether introducing coarse steps is beneficial).
Additionally, the convergence result requires exact gradients at the fine level while using stochastic gradients at the coarse level. The authors acknowledge this gap, but do not explain what technical issue prevents extending the result.

2. Assumption 2 is potentially restrictive. It combines a scaling condition (11) and a growth condition (12). The paper only verifies these in the Euclidean case ($J_\delta = \frac{1}{2}\\|\cdot\\|^2$). It is unclear if it has been used in prior literature, and how restrictive it is in this regime.

3. The claim that ML LinBreg is competitive with or outperforms baselines is overstated for two reasons. First, the paper is using a single fixed learning rate of $0.1$ across all experiments. There is no evidence that the hyperparameters were tuned for any of the algorithms. If no tuning was performed for any method, the comparisons are of very limited value.
Second, even setting aside tuning, the claimed outperformance is not consistently supported by the data. In the majority of experiments the methods are interleaved throughout the sparsity range. An exception is WideResNet28-10 on TinyImageNet, where ML LinBreg does appear to achieve superior accuracy, but this is one case among many. The proposed method is worse than RigL in most experiments. More accurate characterizations would be "comparable to", which the paper itself occasionally uses.

4. The comparison plots show methods at matched sparsity levels. RigL targets a fixed sparsity by design, but ML LinBreg's sparsity comes from the choice of $\lambda$ and cannot be directly controlled. It is therefore not possible to compare all methods at exactly matched sparsity levels. This makes fine-grained comparisons unreliable.

**Requested Changes:**

Critical:

1. Provide an explicit convergence rate, or clearly state that none is available and revise the theoretical claims.
2. Discuss the applicability of Assumption 2 in any non-trivial settings, and whether this assumption has appeared in prior literature. How restrictive is it?
3. Describe the hyperparameter tuning protocol for all methods. If no tuning was performed, the experimental comparisons provide little evidence regarding the relative comparisons of the algorithms.

Would strengthen the work:

4. Soften the "consistently outperforms" statements to reflect what the data shows.
5. Add a more explicit discussion of technical novelty relative to Bauschke et al. (2019) and Elshiaty & Petra (2026), beyond the statement that proofs are "very similar."

---

> ### Author Response · Authors · 2026-07-22
>
> We thank the reviewer for their time and feedback.
> We have revised the manuscript to address their comments (with all major updates highlighted in blue).
> Below, we address their points and questions.
>
>
>
> > *Is there a convergence rate available? What is the advantage of $m > 0$? What about analysis for stochastic gradients on the fine level?*
>
> - Thank you for pointing out the wrong claim of a convergence rate.
> A convergence rate can only be obtained in special cases (for example without stochastic gradients) and we have therefore removed the claim.
>
> - A theoretical advantage of $m > 0$ strongly depends on the quantity $\hat{\rho}_i$. If this is negative, it might be beneficial to use the coarse level from the point of view of our theoretical analysis.
> Empirically, we observe that it is worth setting $m>0$ as is reflected in our numerical experiments. Our method always performs at least as good as the LinBreg algorithm ($m=0$) while offering a potential computational benefit.
>
> - To make a first step in the direction of using stochastic gradients on the fine level, we extended our convergence analysis by including gradient estimators with bounded variance on the fine level.
> In this case, we cannot prove that the expected gap between the achieved loss value and the optimal value vanishes, but we obtain an upper bound in terms of the gradient noise similar to the achieved result for AC/DC (Peste et al. (2021)).
> We would also like to point out that, to the best of our knowledge, there is not even a convergence theory of multilevel stochastic gradient descent which should be much easier to handle than our mirror descent framework.
>
>
>
>
>
> > *Restrictiveness of Assumption 2*
>
> The assumptions we make in Assumption 2 have been introduced in Bauschke et al.(2019).
> They additionally provide some examples, when the assumptions are satisfied.
> For example, the second part of the assumption is fulfilled, if the objective function is strongly convex or if it satisfies a Bregman growth condition.
> Subsequently, Elshiaty and Petra (2026) made the same assumptions in the context of multilevel optimization.
> Also other sparse training approaches like AC/DC rely on an assumption which is similar in flavor.
> We concede that all these assumptions are potentially restrictive and hard to verify, however, we perceive them as a starting point for analyzing mirror-descent-type algorithms.
> Note also that for sparsity regularization locally the assumption is implied by local strong convexity around a minimizer, as argued in Remark 5 of Bungert et al. (2022).
> We added explanations right after Assumption 2.
>
>
>
>
> > *Hyperparameter tuning protocol*
>
> For the choice of hyperparameters, we relied on the information from the papers introducing the respective methods for experiments on the same datasets and architectures.
> Therefore, we used an initial learning rate of $0.1$ for our numerical experiments.
> Moreover, we would like to clarify that we do not fix the learning rate throughout the training but rather use a cosine annealing scheduler to obtain a decaying learning rate.
>
>
>
>
>
> > *Soften the "consistently outperforms" statements to reflect what the data shows*
>
> We have revised several statements throughout the manuscript to ensure that our claims accurately reflect the empirical evidence.
>
>
>
>
>
> > *Differences to Bauschke et al. (2019) and Elshiaty and Petra (2026)*
>
> In the final paragraph of section 3, we briefly discuss the main differences of our approach to ML BPGD (Elshiaty \& Petra (2026)), which is also reflected in the analysis.
> We employ stochastic gradient estimators instead of the exact gradient and use a regularizer that is not differentiable, which necessitates the investigation of the subgradients with the transfer between the levels.
> Moreover, ML BPGD involves a line search and allows for an arbitrary convex function as the coarse objective together with a linear correction term.
> Meanwhile, our specific choice for the coarse objective makes the linear correction term and the line search obsolete.
> However, this requires us to investigate which properties the coarse objective inherits from the fine objective to study the descent during the coarse phases, whereas no such analysis is necessary for ML BPGD.
>
>
>
> > *Symmetrized Bregman divergence is used before its definition, imprecise statement for non-negativity of the Bregman divergence*
>
> We have moved the definition of the symmetrized Bregman divergence to before its first use in Lemma 4.2.
> Moreover, we have adjusted the statement about the non-negativity of the Bregman divergence to point out that this only holds, when $p$ is a subgradient.

---

### Review · Reviewer_g3t4 · 2026-07-07

**Summary Of Contributions:**

The paper proposes ML LinBreg, a dynamic sparse training method built on linearized Bregman iterations (LinBreg). The method runs a full LinBreg update every m+1 iterations. Between these updates, it freezes the current support and updates only the non-zero parameters. The authors view this as a two-level optimization method. The coarse level is the sparse subnetwork selected by a restriction operator, and the fine level is the full parameter space. They then adapt the convergence theory of Multilevel Bregman Proximal Gradient Descent to this setting. The paper contributes the algorithm, a convergence analysis under a Polyak-Lojasiewicz-type condition, and experiments on CIFAR-10 and TinyImageNet with ResNet18, VGG16, and WideResNet28-10. The analysis proves convergence in expectation for the function values, with exact gradients on the fine level and stochastic gradients on the coarse level. It also handles the transfer of subgradients between levels for non-smooth regularizers. The experiments compare against SGD, LinBreg, RigL, and prune-plus-fine-tune. They report a theoretical FLOPs reduction from 0.38x for LinBreg to 0.06x for the proposed method, relative to dense SGD, and about 49% CPU wall-clock savings with SparseProp kernels. The main strengths are the clean optimization view, the careful subgradient argument, and the reasonably broad evaluation. The main weaknesses are that the claimed accuracy gain over LinBreg is not supported at matched sparsity, AC/DC is not compared or clearly distinguished, the strongest convergence statement does not match the experimental regime, and the practical speedup is shown only on CPU.

**Additional Comments:**

Overall, this is a careful paper with a clean optimization-theoretic contribution. Its practical novelty over AC/DC and standard dynamic sparse training is limited, and the real advance is computational rather than accuracy-based. My "No" on the claims question is based on three specific and fixable issues: the accuracy claim versus Table 1, the scope of the theory, and the missing comparison or contrast with AC/DC. If these are addressed, I would support acceptance.

**Audience:**

Yes

**Audience Explanation:**

The paper sits at the intersection of sparse training, mirror descent / Bregman methods, and multilevel optimization. It should interest readers in all three areas. The link between the frozen-support phase and the coarse level of a multilevel method is a useful way to reuse existing convergence tools. The subgradient transfer for non-smooth regularizers is also a real technical contribution. Even readers who are not convinced by the empirical gains may find the framing and analysis useful. The observed kernel-level sparsity, together with the group l_1,2 option, also suggests a direction for future work. TMLR does not require novelty or state-of-the-art results as a strict bar. By the audience-interest criterion, this paper qualifies.

**Claims And Evidence:**

No

**Claims Explanation:**

The well-supported claims are the reported sparsity levels (Table 1), the theoretical FLOPs reduction to 0.06x (Appendix C), and the CPU wall-clock reduction with SparseProp (Figure 3). I have no objection to these claims.

The claims that are not adequately supported:

1. The accuracy claim is contradicted at matched sparsity. Contribution 4 and the text on page 11 say that the method gives sparser models "while maintaining or even improving accuracy" over LinBreg. Table 1 does not show this for ResNet18. ML LinBreg with lambda=0.007 gives 96.12% sparsity and 90.24% accuracy. LinBreg with lambda=0.2 gives 96.03% sparsity and 90.35% accuracy. At almost the same sparsity, the proposed method is slightly less accurate. Prune-plus-fine-tune at 95% sparsity is also more accurate, at 90.84%. VGG16 shows the same pattern. WideResNet28-10 is the only clear win in sparsity, while accuracy is essentially tied (96.58% sparsity at 91.69% versus 95.89% at 91.70%). Given the reported standard deviations of about 0.2 to 0.33, these accuracy gaps look like noise. The fair reading is accuracy parity with LinBreg, not improvement. The claim should be reframed.
2. The convergence "guarantees" hold in a setting that the experiments do not use. Theorem 4.4 assumes exact gradients on the fine level, but the experiments use minibatch gradients. Corollary 4.5 gives exact convergence to the optimum only when the coarse step sizes go to zero. In that limit, the frozen-support coarse phase, which is the part that saves compute, becomes asymptotically inactive. Thus the cleanest guarantee applies in a limit where the method behaves like plain LinBreg. In practice, the experiments use a fixed cosine schedule with m=99. This is closer to the constant-step setting, where Theorem 4.4 gives only a noise-neighborhood result. The paper should state this gap clearly. The abstract claim that "we provide convergence guarantees" currently reads stronger than what is shown for the algorithm as run.
3. The improvement over prior work is not benchmarked against the closest method. AC/DC (Peste et al., NeurIPS 2021) alternates dense phases with sparse phases. In the sparse phases, the mask is frozen and only non-zero weights are updated. It also proves convergence for an iterative-hard-thresholding variant under the same PL condition used here. This is very close to the full-update / frozen-support structure in this paper. Without an empirical comparison, or at least a clear conceptual distinction, the claimed improvement over existing sparse training is not convincing.

If the authors reframe the accuracy result as parity, make the theory-practice gap explicit, and either compare to or clearly distinguish from AC/DC, I would change my answer to Yes.

**Requested Changes:**

* Reconcile the accuracy claims with Table 1. At matched sparsity ML LinBreg is not more accurate than LinBreg for ResNet18 and VGG16, and pruning is often best on accuracy. Please reframe contribution 4 and the page-11 text around the computational savings, which are the genuine advance, and state the accuracy result as parity. Reporting significance (or noting that gaps are within the standard deviations) would help.
* State the scope of the convergence theory clearly near Theorem 4.4 and Corollary 4.5. The guarantee assumes exact fine-level gradients, which the minibatch experiments do not satisfy. Exact convergence also requires vanishing coarse step sizes, while the experiments use constant steps. The abstract phrase "we provide convergence guarantees" should be softened to match this scope.
* Add a paragraph that distinguishes the method from AC/DC (Peste et al., 2021). AC/DC has the same alternating full-update / frozen-support structure and also uses a PL-based convergence argument. Please explain what the Bregman / multilevel derivation adds beyond iterative hard thresholding under PL.
* Empirically compare to AC/DC and to at least one modern dynamic-sparse-training method at extreme sparsity, such as Top-KAST (Jayakumar et al., 2021), MEST (Yuan et al., 2021), or Ji et al. (ICML 2024). TMLR does not require exhaustive baselines. Still, showing parity with a recent method on one architecture and one dataset would strengthen the paper. "Comparable to RigL (2020)" is a weak position in 2026.
* Clarify the FLOPs accounting in Appendix C. The full-update step is counted as 2f_S + f_D. However, a full LinBreg update needs gradients for all parameters, so the dense backward pass seems closer to 2f_D than f_D. Since the 0.06x headline number depends on this, please confirm or correct the formula.
* Add a convergence plot under the actual stochastic setting, with minibatch fine gradients and constant steps. This would show whether the practical behavior matches the theory qualitatively.
* Push the emergent structured, kernel-level sparsity toward a GPU wall-clock result. The S_conv analysis and the group l_1,2 regularizer are promising. However, the current speedup is CPU-only and uses a specialized kernel. The Torch version is actually slower than SGD (Figure 3). A GPU-realizable speedup would make the practical case much stronger.

---

> ### Author Response · Authors · 2026-07-22
>
> We thank the reviewer for their time and feedback.
> We have revised the manuscript to address their comments (with all major updates highlighted in blue).
> Below, we address their points and questions.
>
>
>
> > *Reframe contribution 4 and the page-11 text around the computational savings, which are the genuine advance, and state the accuracy result as parity*
>
> We agree that, for high sparsity levels on CIFAR-10, our results indicate that the proposed multilevel approach achieves performance comparable to the standard LinBreg algorithm rather than a clear improvement.
> However, we would like to emphasize that on TinyImageNet, the multilevel approach consistently demonstrates improved performance (see Figure 4).
> We have therefore revised our claims accordingly to more accurately reflect these findings.
>
>
>
>
>
> > *Scope of convergence theory*
>
> We agree on the discrepancy between the theoretical results and the numerical experiments.
> To make a first step in this direction, we added stochastic gradients with bounded variance to the analysis in the updated version of our manuscript.
> In this case, we cannot prove that the expected gap between the achieved loss value and the optimal value vanishes, but we obtain an upper bound in terms of the gradient noise similar to the result for AC/DC (Peste et al. (2021)).
> We would also like to point out that, to the best of our knowledge, there is not even a convergence theory of multilevel stochastic gradient descent which should be much easier to handle than our mirror descent framework.
>
>
>
>
>
>
> > *Distinguish from AC/DC*
>
> We have added a paragraph to discuss the connections of our proposed method to the AC/DC algorithm at the end of section 3.
> The main conceptual difference is that AC/DC discards all information about parameters that are not selected for the compressed phase whereas
> our algorithm retains such previous information.
> However, with some minor changes, our algorithm can recover an AC/DC type algorithm.
>
> Moreover, we are currently working on numerical comparisons to AC/DC and MEST (Yuan et al. (2021) and will include them as soon as the results are available.
>
>
>
>
> > *FLOP counting formula*
>
> For comparability, we followed the computation from Evci et al. (2020).
> They argue that the forward pass and back-propagating the error signal takes $2 f_S$ FLOPs, while calculating the dense gradients requires another $f_D$ FLOPs, leading to a total number of $2 f_S + f_D$ FLOPs for calculating the full gradients.
>
>
>
>
>
> > *Add a convergence plot under the actual stochastic setting, with minibatch fine gradients and constant steps*
>
> Since we have extended our theoretical analysis to cover minibatch fine-level gradients, the training loss plot shown in Figure 6 of the Appendix now matches the setting considered in our theory in terms of the use of minibatch gradients.
>
> > *Push the emergent structured, kernel-level sparsity toward a GPU wall-clock result*
>
> We agree that exploiting structured sparsity represents an important avenue for future research.
> However, we believe that investigating this and developing custom kernels is beyond the scope of the present work.

---

### Review · Reviewer_Hj63 · 2026-07-12

**Summary Of Contributions:**

This paper proposes Multilevel LinBreg, a dynamic sparse training method based on linearized Bregman iterations / mirror descent. The method periodically restricts updates to the currently selected parameters during coarse-level steps and returns to the full parameter space so that the support can change. The authors formulate this procedure as a multilevel optimization method, analyze subgradient transfer between the coarse and fine levels, and provide convergence analysis for a mixed-gradient setting. Experiments on CIFAR-10 and TinyImageNet show competitive sparsity-accuracy tradeoffs against several relevant baselines, as well as promising computational savings in the reported settings.

### Strengths

1. By periodically restricting the coarse-level updates to the currently selected parameters for several consecutive coarse-level steps, the method exploits the sparsity induced by LinBreg to reduce the required gradient information. The design is conceptually simple and has a clear computational motivation.

2. The authors formulate the proposed algorithm as a multilevel optimization method using dynamic restriction and prolongation operators. They also analyze the transfer of subgradients between the coarse and fine levels, which is a useful technical component of the multilevel formulation.

3. The paper evaluates the method on two datasets, multiple network architectures, and several relevant baselines. It also includes ablation studies on the regularization parameter, the duration of the coarse period, and the mask initialization scheme. The results suggest that the method can improve the sparsity-accuracy tradeoff of standard LinBreg in several settings. The theoretical FLOP analysis and the sparsity-aware CPU implementation also provide preliminary evidence of potential computational savings in the tested settings.

### Weaknesses

1. The main convergence theorem considers exact gradients at the fine level and stochastic gradients at the coarse level, whereas all updates in the experiments use mini-batch gradients. The current theory therefore does not directly cover the experimental implementation.

2. Assumption 2 is fundamental to the main convergence result, but the paper does not sufficiently explain when this condition holds in the sparse training setting considered in the experiments. Representative sparse-training cases or a discussion of its limitations would help readers better understand the connection between the theory and the practical sparsification mechanism.

3. The paper involves several levels of notation, definitions, and interdependent assumptions. The authors should carefully recheck the mathematical presentation, correct typographical errors, and clearly state the conditions required or inherited by each theorem. For example, the step-size condition at the end of the derivation of Lemma B.2 may need to be rechecked for its dependence on $L$, including the corresponding statement in Theorem B.3.

4. The current experiments focus on convolutional architectures. Evaluation on a different model family would provide additional evidence for broader applicability, while a substantially larger-scale setting would provide complementary evidence on scalability.

**Audience:**

Yes

**Audience Explanation:**

This paper should interest researchers in dynamic sparse training, Bregman methods, and efficient deep learning. Periodically restricting support expansion during coarse-level updates is intuitive and potentially practical, while the multilevel optimization view differs from standard pruning-regrowth methods. Although the experiments could be expanded, the current results indicate that the proposed direction is relevant to this community.

**Claims And Evidence:**

No

**Claims Explanation:**

The main empirical findings are generally supported by the reported results within the tested settings. I selected No overall because some of the theoretical claims and the scope of the broader conclusions still require more precise qualification, as discussed in the weaknesses part. These concerns appear addressable through revised wording, additional clarification, targeted mathematical corrections, and a clearer delimitation of the empirical scope, and they do not undermine the main empirical findings of the paper.

**Requested Changes:**

1. Clearly distinguish the mixed-gradient algorithm covered by the theoretical analysis from the fully mini-batch implementation used in the experiments, and revise the convergence claims in the abstract, introduction, and conclusion accordingly.

2. Further clarify the applicability of Assumption 2 to the sparse training setting considered in this paper, for example through representative cases or a clearer statement of its limitations.

3. Recheck and correct typographical errors, dependencies among assumptions, and the step-size and notation issues in the mathematical analysis.

4. The empirical evaluation could be further strengthened by including at least one more recent dynamic sparse training method as a baseline. An experiment on a different model family or at a substantially larger scale than the current settings would also be useful for assessing broader applicability beyond CNN-based image classification.

---

> ### Author Response · Authors · 2026-07-22
>
> We thank the reviewer for their time and feedback.
> We have revised the manuscript to address their comments (with all major updates highlighted in blue).
> Below, we address their points and questions.
>
>
>
>
> > *Convergence result uses exact gradients for fine level, while experiments use mini-batch approximations*
>
> We agree on the discrepancy between the theoretical results and the numerical experiments.
> To address this issue, we extended our convergence analysis by including stochastic gradients with bounded variance for the steps on the fine level in the updated version of our manuscript.
> In this case, we cannot prove that the expected gap between the achieved loss value and the optimal value vanishes, but we obtain an upper bound in terms of the gradient noise similar to the result for AC/DC (Peste et al. (2021)).
> We would also like to point out that, to the best of our knowledge, there is not even a convergence theory of multilevel stochastic gradient descent which should be much easier to handle than our mirror descent framework.
>
>
>
>
>
> > *Restrictiveness of Assumption 2*
>
> The assumptions we make in Assumption 2 have been introduced in Bauschke et al. (2019).
> They additionally provide some examples, when the assumptions are satisfied.
> For example, the second part of the assumption is fulfilled for strongly convex objective functions or for objective functions satisfying a Bregman growth condition.
> Subsequently, Elshiaty and Petra (2026) made the same assumptions in the context of multilevel optimization.
> Also other sparse training approaches like AC/DC rely on an assumption which is similar in flavor.
> We concede that all these assumptions are potentially restrictive and hard to verify, however, we perceive them as a starting point for analyzing mirror-descent-type algorithms.
> Note also that for sparsity regularization locally the assumption is implied by local strong convexity around a minimizer, as argued in Remark 5 of Bungert et al. (2022).
> We added explanations right after Assumption 2.
>
>
>
>
> > *Typographical errors and mathematical presentation*
>
> Thank you for pointing us to the missing dependence on $L$  of the step-size bounds in Appendix B.
> We have corrected this typo in the new version of our manuscript.
>
>
>
>
>
> > *Different model family for experiments*
>
> We are currently running experiments for ViT-B-16 (Dosovitskiy et al. (2021)), a transformer-based architecture, and will provide results as soon as they are available.